# Mitochondrial DNA removal is essential for sperm development and activity

Zhe Chen[1], Fan Zhang [1], Annie Lee[1], Michaela Yamine[1], Zong-Heng Wang[1], Guofeng Zhang[2], Christian Combs [1] & Hong Xu [1✉]

## Abstract

**Active mitochondrial DNA (mtDNA) elimination during spermatogenesis has emerged as a conserved mechanism ensuring the uniparental mitochondrial inheritance in animals. However, given the existence of post-fertilization processes degrading sperm mitochondria, the physiological significance of mtDNA removal during spermatogenesis is not clear. Here we show that mtDNA clearance is indispensable for sperm development and activity. We uncover a previously unappreciated role of Poldip2 as a mitochondrial exonuclease that is specifically expressed in late spermatogenesis and required for sperm mtDNA elimination in *Drosophila*. Loss of Poldip2 impairs mtDNA clearance in elongated spermatids and impedes the progression of individualization complexes that strip away cytoplasmic materials and organelles. Over time, *poldip2* mutant sperm exhibit marked nuclear genome fragmentation, and the flies become completely sterile. Notably, these phenotypes were rescued by expressing a mitochondrially targeted bacterial exonuclease, which ectopically removes mtDNA. Our work illustrates the developmental necessity of mtDNA clearance for effective cytoplasm removal at the end of spermatid morphogenesis, and for preventing potential nuclear-mitochondrial genome imbalance in mature sperm, in which nuclear genome activity is shut down.**

**Keywords** Maternal Inheritance; Exonuclease; *Drosophila* spermatogenesis; Male Sterile; EndoG
**Subject Categories** Development; DNA Replication, Recombination & Repair; Post-translational Modifications & Proteolysis

See also: Z Wang et al

## Introduction

The mitochondrial genome is transmitted exclusively through the maternal lineage in most sexually reproduced organisms. However, the underlying mechanisms of uniparental inheritance are not well-understood, and its physiological significance remains elusive. Maternal inheritance was once attributed to the simple dilution of sperm mitochondrial DNA (mtDNA) in the zygote, which contains an enormous amount of maternal mtDNA (Gyllensten et al, 1991; Wolff and Gemmell, 2008). However, increasing evidence suggests that mtDNA and mitochondria-derived vesicles are actively eliminated during spermatogenesis (Nishimura et al, 2006; DeLuca and O'Farrell, 2012; Yu et al, 2017; Lee et al, 2023) or embryogenesis (Sutovsky et al, 1999; Sutovsky et al, 2000; Zhou et al, 2016; Politi et al, 2014; Sato and Sato, 2011), respectively. Considering the high energy demand of the spermatogenesis process and sperm motility, it is intriguing why fathers proactively remove sperm mtDNA before fertilization. Understanding the evolutionary drive of mtDNA removal in spermatogenesis is of great interest. In *Drosophila* testis, abundant mitochondrial nucleoids are observed along the length of mitochondrial derivatives in elongating spermatids (DeLuca and O'Farrell, 2012), but the majority of them abruptly disappear in fully elongated spermatids. Few remaining nucleoids are stripped away by progressing actin cones during spermatid individualization and eventually end up in waste bags. EndoG (DeLuca and O'Farrell, 2012), a mitochondrial endonuclease, and Tamas (Yu et al, 2017), the mitochondrial DNA polymerase, are involved in the pre-individualization mtDNA removal. EndoG is a site-specific endonuclease, targeting both mtDNA (Cote and Ruizcarrillo, 1993; Ruizcarrillo and Renaud, 1987) and nuclear DNA (nuDNA) (Li et al, 2001). Notably, the polymerase activity of Tamas, not its exonuclease activity, is required for mtDNA removal (Yu et al, 2017). Hence, additional nucleases, particularly an exonuclease, are likely involved in further degrading EndoG-nicked mtDNA. In addition, the pleiotropy of EndoG and Tamas poses challenges to understanding the physiological significance of mtDNA removal during spermatogenesis.

## Results

### A testis mitochondrial nucleoid protein, Poldip2, is required for male fertility

We hypothesized that a protein specifically involved in mtDNA removal would likely show biased expression in the testis and be associated with mitochondrial nucleoids. To this end, we surveyed the expression patterns (Leader et al, 2018; Roy et al, 2010) of *Drosophila* homologs of previously identified mammalian nucleoid proteins (Han

[1]National Heart, Lung, and Blood Institute, NIH, Bethesda, MD, USA. [2]National Institute of Biomedical Imaging and Bioengineering, NIH, Bethesda, MD, USA.
✉E-mail: hong.xu@nih.gov

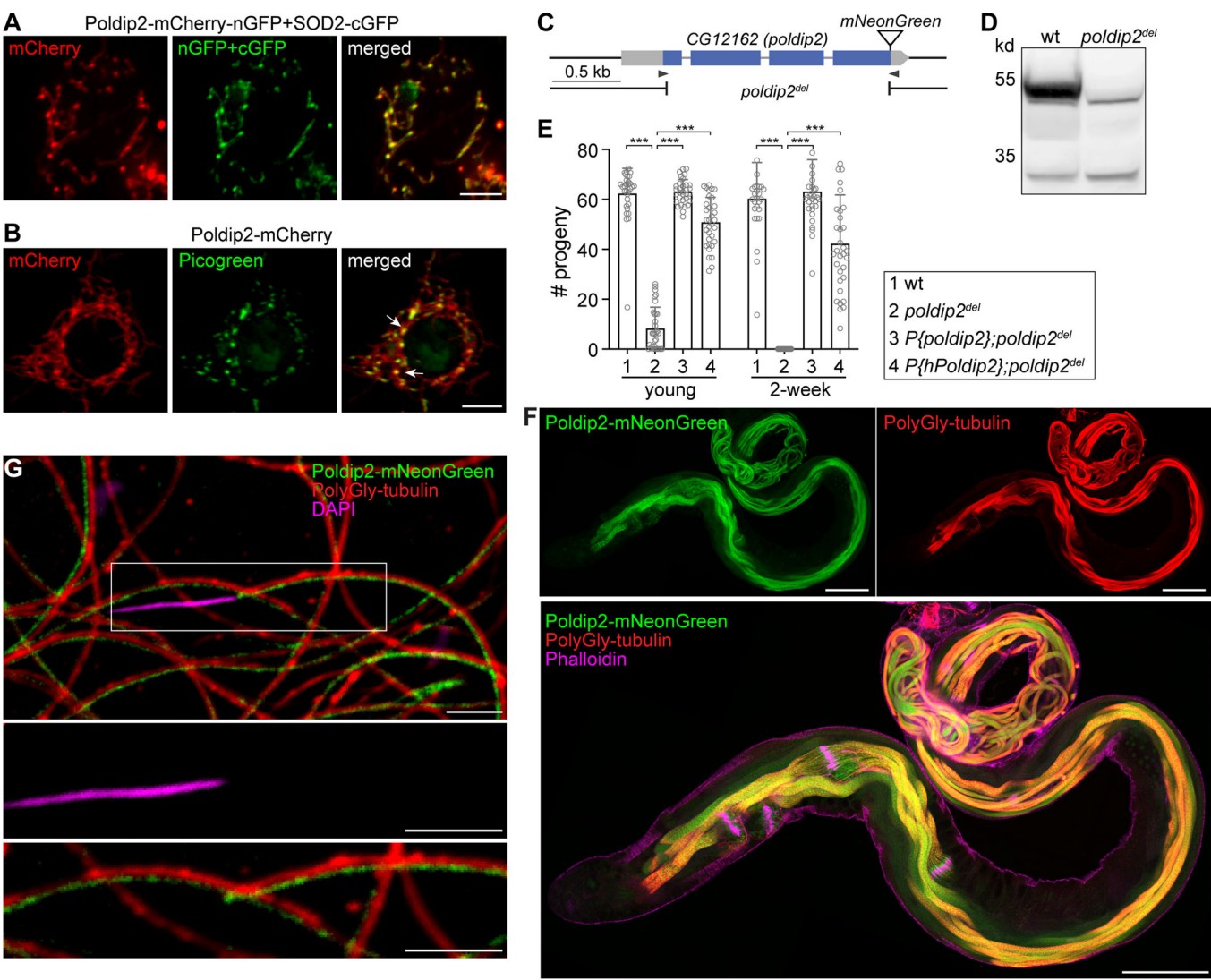

**Figure 1.** A mitochondrial nucleoid protein, Poldip2, is required for male fertility.

(A) A representative image of an S2 cell co-expressing Poldip2-mCherry-nGFP (red) and SOD2-cGFP. Two half-GFP molecules reconstitute into a functional whole GFP (green), demonstrating that Poldip2 co-localizes with SOD2 in the mitochondrial matrix. Bar, 5 µm. (B) Poldip2 concentrates on specific loci on mitochondria, which stain positively for Picogreen (arrows), indicating Poldip2 is associated with mitochondrial nucleoids. Bar, 5 µm. (C) Schematic representation of the *CG12162* genomic locus, illustrating *CG12162* transcripts, 5′- and 3′-UTR (gray bars), exons (blue bars), and the deleted region in *poldip2del*. Arrowheads illustrate the target sites of guide RNAs used for generating the *poldip2* deletion, and for knocking in *mNeonGreen*. (D) Western blot confirms the deletion of the Poldip2 protein in *poldip2del* flies. (E) The *poldip2del* flies are male semi-sterile. The number of progeny produced per day is shown. Compared with wild-type, male *poldip2del* flies produce significantly fewer progeny at a young age and become completely sterile after two weeks. The semi-sterile phenotype can be rescued by expressing either Poldip2 or hPoldip2 protein in *poldip2del* flies ($n \geq 30$). The data represent the mean± SD. Statistical analysis was performed using an unpaired $t$ test. $P$ values from left to right: ***$P = 1.5 \times 10^{-31}$, $P = 8.0 \times 10^{-40}$, $P = 4.5 \times 10^{-27}$, $P = 4.2 \times 10^{-34}$, $P = 7.1 \times 10^{-38}$, $P = 1.4 \times 10^{-18}$. (F) Poldip2 is highly expressed in fully elongated spermatids and all subsequent developmental stages within *Drosophila* testes. Polyglycylated tubulin marks fully elongated axonemal microtubules. Phalloidin stains actin cones and outlines the testis. Bar, 100 µm. (G) Representative image showing that Poldip2 aligns alongside microtubules and is absent from the nuclear head region in spermatozoa, demonstrating that Poldip2 exclusively localizes in mitochondria in mature spermatozoa (enlarged view outlined). Green: Poldip2-mNeon-Green; Red: Polyglycylated tubulin; Magenta: DAPI; bar, 5 µm. Source data are available online for this figure.

et al, 2017; Bogenhagen et al, 2008; Rajala et al, 2015) (Fig. EV1; Dataset EV1). Among them, a candidate nucleoid protein encoded by the *CG12162* locus, known as Poldip2, was found to be highly enriched in testes (Fig. EV1). Poldip2 was initially identified as a human polymerase δ P50-interacting protein in a yeast two-hybrid screen (Liu et al, 2003) and has been proposed to be involved in nuclear genome replication and repair (Maga et al, 2013). However, Poldip2 was exclusively localized to the mitochondrial matrix based on a GFP complementation assay (Fig. 1A) and concentrated on mitochondrial nucleoids (Fig. 1B), suggesting its potential association with mitochondrial DNA in testes.

Using CRISPR/Cas9-mediated non-homologous end joining, we generated a deletion, *poldip2del*, which removed most of the coding region of *poldip2* (Fig. 1C). No Poldip2 protein was detected in

*poldip2^del* flies (Fig. 1D), indicating that this deletion is a null allele. The *poldip2^del* flies were largely healthy, except that male flies were semi-sterile (Fig. 1E). Newly-eclosed male *poldip2^del* flies produced much fewer progeny compared to wild-type (wt) flies and became completely sterile after two weeks (Fig. 1E). The fertility of *poldip2^del* male flies was fully restored by a minigene spanning the genomic region of *poldip2* (Fig. 1E), demonstrating that impaired male fertility is caused by the loss of Poldip2. Additionally, a transgene containing the cDNA of the human homolog of *poldip2*, *hPoldip2*, flanked by the 5′- and 3′-UTRs of *poldip2*, largely restored the fertility of *poldip2^del* male flies, indicating a conserved function of Poldip2 in maintaining male fertility in metazoans (Fig. 1E).

## Persistent mtDNA impairs male fertility

To elucidate the underlying cause of the impaired fertility of *poldip2^del* male flies, we examined the developmental expression pattern of Poldip2 in the testes. We generated a Poldip2-mNeonGreen (Poldip2-mNG) reporter by inserting the *mNeon-Green* cDNA into the endogenous locus of *poldip2* (Fig. 1C). Poldip2-mNG expression was hardly detected in early spermatogenesis stages but exhibited an abrupt increase in fully elongated spermatids, which were marked by polyglycylated tubulin (Fig. 1F). Its expression persisted through all subsequent developmental stages, including mature spermatozoa deposited in seminal vesicles (Fig. 1G).

The onset of Poldip2 expression coincided with mtDNA removal in elongated spermatids (DeLuca and O'Farrell, 2012). In addition, Poldip2 was found to be enriched on mitochondrial nucleoids (Fig. 1B). These observations led us to test whether it is involved in mtDNA removal. We stained isolated spermatid bundles with DAPI, a fluorescent DNA dye, to visualize mtDNA. In elongating spermatids, mitochondrial nucleoids were evenly distributed in both wt and *poldip2^del* sperm tails (Fig. 2A). The total number of nucleoids in early- and mid-elongating spermatids was comparable between wt and *poldip2^del* (Figs. 2D and EV2A), although some nucleoids were notably larger in *poldip2^del* spermatids (Fig. EV2C), suggesting a potential defect in nucleoid organization. In fully elongated spermatids, most mitochondrial nucleoids disappeared in wt (Figs. 2B,D,E and Fig. EV2A,B), indicating active mtDNA removal at this stage. In contrast, a significant amount of mtDNA remained in *poldip2^del* spermatids (Figs. 2B,D,E and EV2A,B) and persisted after individualization (Fig. 2C). Since intact seminal vesicles are impermeable to DAPI, we imaged an endogenously tagged TFAM reporter, TFAM-mNeonGreen (Zhang et al, 2024), which marks mitochondrial nucleoids (Fig. EV2D,E), to assess mtDNA in mature sperm. Many TFAM puncta were detected in mature sperm in both young and 2-week-old *poldip2^del* seminal vesicles, while no nucleoids were found in wt sperm (Figs. 2F and EV2F).

We further quantified the copy number of mtDNA molecules in mature sperm. Either *w^1118* or *poldip2^del* male flies carrying wild-type mtDNA (*mt:wt*) were mated with *w^1118* female flies carrying homoplasmic *mt:ND2^del1*, which contains a 9-base pair deletion on mtDNA-encoded *ND2* locus (Xu, 2008). Subsequently, we dissected the sperm storage organ, the spermathecae, from copulated female flies and subjected them to droplet digital PCR analysis, using primers specifically targeting paternal mtDNA

(*mt:wt*) (Fig. EV2G,G'). In crosses using *w^1118* male flies, no paternal mtDNA was detected (Fig. EV2H,H'), supporting the notion that mature sperm are devoid of mtDNA in *Drosophila* (DeLuca and O'Farrell, 2012) (Yu et al, 2017). However, when the fathers were *poldip2^del* flies, a significant amount of paternal mtDNA was detected (Fig. EV2H,H'). On average, each mature sperm from *poldip2^del* contained approximately 60 copies of mtDNA (Fig. 2G). The *poldip2^del* sperm fertilized significantly fewer embryos compared to wt sperm (Appendix Fig. S1A–C). However, they could initiate embryonic development once successful fertilization occurred (Appendix Fig. S1B). Notably, paternal mtDNA from *poldip2^del* sperm was detected in embryos 30 min after egg-laying but disappeared after 6 h (Figs. 2H and EV2I,I'). When ATG7, a key enzyme involved in initiating the autophagy pathway, was maternally knocked down, more paternal mtDNA was detected in embryos but was eventually eliminated (Appendix Fig. S1D), consistent with a previous study showing that sperm mitochondria are degraded during early embryogenesis (Politi et al, 2014). Altogether, these observations demonstrate that Poldip2 is essential for mtDNA removal during late spermatogenesis.

We next addressed whether the persistence of mtDNA in mature sperm is the cause of impaired fertility in *poldip2^del* male flies. Given that EndoG nicks mtDNA in developing spermatids (DeLuca and O'Farrell, 2012), we reasoned that ectopically introducing an exonuclease into mitochondria would degrade mtDNA nicked by EndoG in *poldip2^del* spermatids. We replaced the coding region of *poldip2* with a fusion gene consisting of a mitochondrial-targeting sequence and the cDNA encoding *Escherichia coli* (*E.coli*) Exonuclease III (*mitoExoIII*), using CRISPR/Cas9-mediated recombination (Fig. 3A). To prevent potential leaky expression of mitoExoIII, we placed the SV40 transcription-terminating sequence (Connelly and Manley, 1988), flanked by two Flippase (FLP) recombination target (FRT) sites, in front of *mitoExoIII*. In the presence of FLP activated by the *Bam-gal4 driver*, FRT-mediated recombination excises the termination sequence, allowing the expression of mitoExoIII under the control of the *poldip2* promoter exclusively in the germline. Expression of mitoExoIII in the heteroallelic combination of the *poldip2* null background reduced the abundance of remaining mitochondrial nucleoids in mature sperm (Fig. 3B), and importantly, restored male fertility in both young and old *poldip2* null flies (Fig. 3C), indicating that the persistence of mtDNA in late spermatogenesis impairs male fertility.

## Poldip2 is a mitochondrial exonuclease

A previous study showed that Poldip2 diminished the signal of DNA probes (Maga et al, 2013), although this phenomenon has not been further investigated. Herein, we found that a foreign exonuclease can functionally replace Poldip2 in *Drosophila*. Additionally, Poldip2 exhibits moderate structural similarity to mammalian Dom3Z (Appendix Fig. S3), an RNA nuclease with 5′-3′ exoribonuclease activity (Jiao et al, 2013). These observations prompted us to explore whether Poldip2 might have nuclease activity. We expressed and purified the full-length recombinant Poldip2 protein from *E.coli* to a purity above 96% (Fig. EV3A). Incubation of 5′-6-FAM-labeled 20-nt poly(dT) (Fig. 3D) or poly(dA) (Fig. EV3B) with recombinant Poldip2 resulted in a

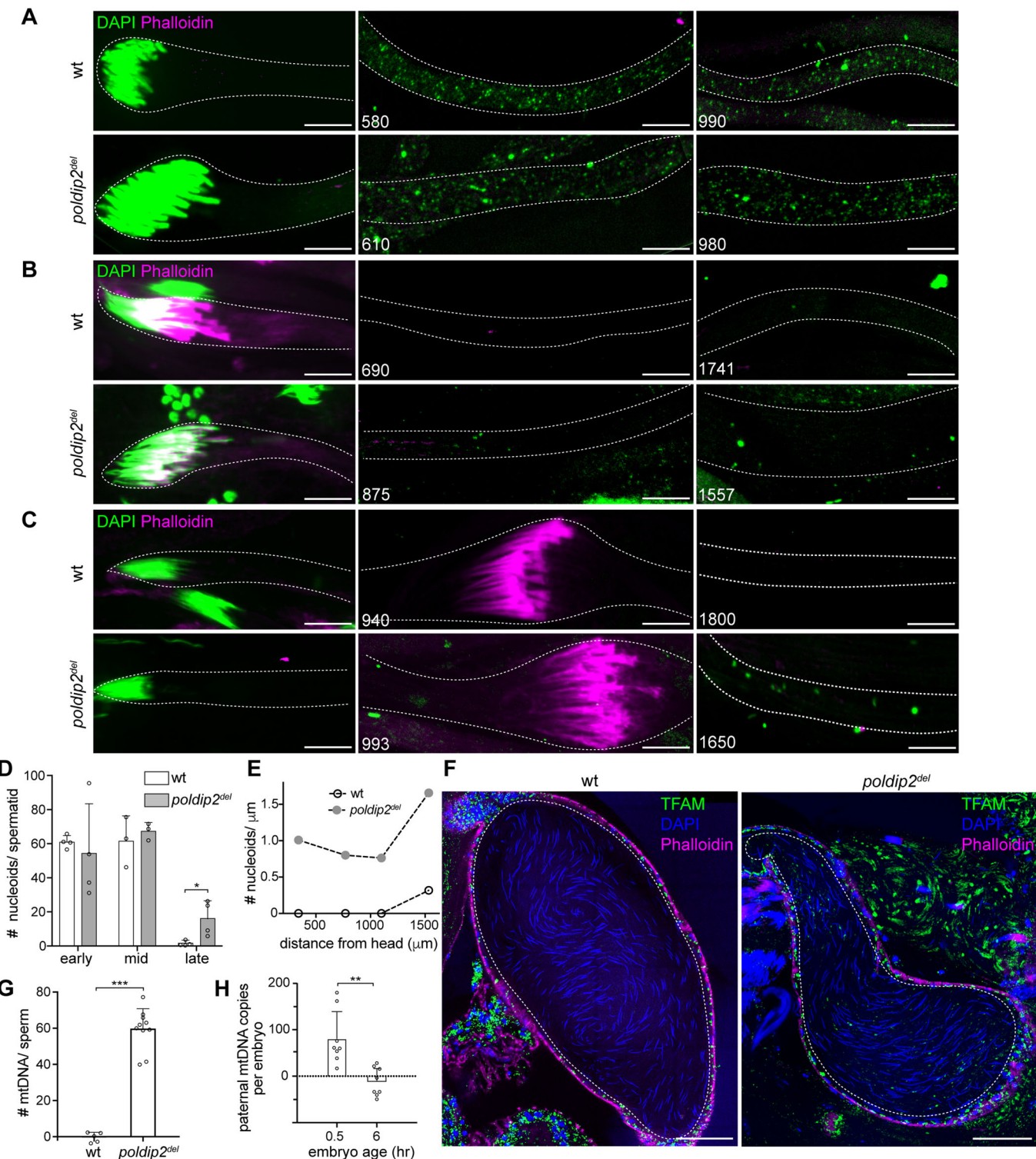

ladder-like pattern of oligonucleotides ranging from monomer to 19-mer. Poldip2 also degraded 3′-6-FAM-labeled 20-nt poly(dT) (Fig. 3E) or poly(dA) (Fig. EV3C) in the 5′-3′ direction. In a reaction using a single-stranded DNA (ssDNA) substrate consisting of mixed nucleotides (Fig. EV3D), different intensities of degradation intermediates were observed, suggesting a potential nucleotide

preference of Poldip2. Indeed, Poldip2 showed minimal degradation of a 20-nt poly(dC) (Fig. EV3E–H), and a stretch of tandem dC hindered the progression of Poldip2 on a ssDNA substrate (Fig. 3F). Furthermore, we examined Poldip2's exonuclease activity on double-stranded DNA (dsDNA) substrates with various configurations. While the intact dsDNA substrate exhibited minimal

◄

**Figure 2.   Mitochondrial DNA persists in the late stages of spermatogenesis in *poldip2del* flies.**

(A–C) Representative images of elongating (A), fully elongated (B), and individualization (C) spermatid bundles isolated from $w^{1118}$ (wt) and *poldip2del* flies, and stained for DNA (DAPI, green) and actin cones (Phalloidin, magenta). Note that DAPI stains both nuclear DNA (nuDNA) and mitochondrial DNA (mtDNA) in isolated spermatid bundles. Dashed lines outline bundles. Numbers indicate the distance (μm) from the nuclear head. Bar, 10 μm. (D) Quantification of the total mitochondrial nucleoid numbers per spermatid at early-elongating (early), mid-elongating (mid) and fully elongated (late) stages. Each data point represents a spermatid ($n = 3, 4$). The data represent the mean± SD. Statistical analysis was performed using an unpaired $t$ test. *$P = 0.029$. (E) The density of mitochondrial nucleoids (total numbers per μm) along the length of representative fully elongated spermatid bundles for $w^{1118}$ (wt) and *poldip2del* flies, respectively. (F) Representative images showing many mitochondrial nucleoids labeled by TFAM-mNeonGreen in *poldip2del*, but not in wt mature sperm (dashed lines). Phalloidin stains actin (magenta); DAPI stains needle-shaped nuDNA (blue). Bar, 50 μm. (G) Droplet digital PCR (ddPCR) quantification of the average number of paternal mtDNA molecules per sperm in the female spermatheca. Crosses were performed between female $w^{1118}$ ($mt:ND2^{del1}$) flies and male $w^{1118}$ ($mt:wt$) or *poldip2del* ($mt:wt$) flies. Each data point represents a biological replicate ($n = 5, 10$). The data represent the mean± SD. Statistical analysis was performed using an unpaired $t$ test. ***$P = 3.8 \times 10^{-8}$. See also Fig. EV2H,H′. (H) Droplet digital PCR (ddPCR) quantification of the *poldip2del* sperm-derived mtDNA in embryos. Crosses were conducted between female $w^{1118}$ ($mt:ND2^{del1}$) and male *poldip2del* ($mt:wt$) flies. Embryos were collected 0–30 min post-laying and analyzed immediately (0.5 h, $n = 8$) or after 6 h (6 h, $n = 8$) of development. Crosses between female $w^{1118}$ ($mt:ND2^{del1}$) and male $w^{1118}$ ($mt:wt$) were used as the negative control, as no sperm mtDNA is expected in this cross. Each data point represents a biological replicate ($n = 8$). The data represent the mean ± SD. Statistical analysis was performed using an unpaired $t$ test. **$P = 0.0014$. See also Fig. EV2I,I′. Source data are available online for this figure.

degradation by Poldip2, dsDNA with either a nick or a gap was susceptible to degradation (Fig. 3G). Collectively, these results demonstrate that Poldip2 is a DNA exonuclease, which can degrade both ssDNA and dsDNA with breaks and prefers dA/dT over dG/dC. Given the distinct nuclease activity observed in these experiments, the likelihood of bacterial protein contamination is considered low, although it cannot be entirely excluded.

Poldip2 is highly enriched in testes (Leader et al, 2018; Roy et al, 2010). To further explore its role in mtDNA degradation, we overexpressed Poldip2 in several other cell types with lower endogenous expression and examined its effect on mtDNA levels. Ectopic expression of Poldip2 markedly decreased the number of mitochondrial nucleoids in S2 cells (Appendix Fig. S2A) and ovarian germ cells (Appendix Fig. S2B), but not in terminally differentiated midgut enterocytes (Appendix Fig. S2C). Notably, mtDNA replication is active in both S2 cells and developing oocytes (Hill et al, 2014; Zhang et al, 2016), which may generate mtDNA breaks (Goddard and Wolstenholme, 1978; Joers and Jacobs, 2013), making these DNA molecules more susceptible to Poldip2 degradation.

## Persistent mtDNA impedes individualization

We have established that Poldip2 is a mitochondrial exonuclease highly enriched in the testes, specifically degrading mtDNA in elongated spermatids. We next explored how the presence of mtDNA impairs male fertility in *poldip2del* flies. The testes of *poldip2del* flies did not exhibit obvious morphological defects (Fig. EV4A,F). Consistent with Poldip2's spatial pattern, early stages of spermatogenesis seemed unaffected in *poldip2del* testes. While the formation of individualization complexes (ICs), traveling ICs and waste bags also appeared normal in *poldip2del* testes (Fig. EV4B–D,G–I), the coiling region was notably enlarged (Figs. 4A and  EV4K,L), accumulating many needle-shaped nuclei, some of which remained bundled together (Fig. 4A, inset), suggesting a potential individualization defect. The seminal vesicles of both 3-day and 2-week-old *poldip2del* flies were notably smaller and contained fewer mature sperm compared to wt (Figs. 4B and EV4E,J). In transmission electron microscopy analysis of the individualized cyst of *poldip2del* flies, some spermatids were still enveloped in a continuous membrane structure, aside from the remaining spermatids that were properly individualized (Fig. 4C, red arrowhead). In addition, individualized spermatids showing incomplete membrane contours were frequently observed (Fig. 4C,

red arrows). Both phenotypes suggest a potential defect in sperm individualization.

If persistent mtDNA impedes ICs progression, one would expect that a greater amount of remaining mtDNA would lead to a stronger individualization defect. Poldip2, a mitochondrial exonuclease, might work in synergy with EndoG, a mitochondrial endonuclease, to rapidly eliminate mtDNA in elongated spermatids (DeLuca and O'Farrell, 2012). Hence, we attempted to examine the spermatid individualization in a background lacking both *poldip2* and *endoG*. We deleted the entire coding region of *endoG* using CRISPR technology and combined *poldip2del* with a trans-heterozygous combination consisting of *endoG^{KO} and EndoG^{MB07150}*. The double mutant (*endoG^{MB07150/KO}; poldip2del*) was completely sterile, in contrast to the normal fertility of trans *endoG* (*endoG^{MB07150/KO}*) (Fig. 4D). Individual nucleoids were further enlarged on average (Figs. 4E–G and  EV4M), indicating more severe defects in nucleoid organization or morphology. In elongated spermatids, ~74.5% of mitochondrial nucleoids persisted in the double mutant, whereas 11.3% and 23.4% remained in the *endoG* mutant and *poldip2del*, respectively (Fig. 4H). Importantly, the traveling ICs were disorganized, and abundant mtDNA was detected at both basal and distal regions of cystic bulges in the double mutant (Fig. 4G). The observation that more remaining mtDNA caused more severe individualization defects further substantiates the necessity of mtDNA removal for individualization, allowing rapid and smooth progress of traveling ICs.

## Persistent mtDNA may impact nuclear genome integrity

Having demonstrated that mtDNA removal promotes sperm individualization, which explains the reduced fertility of young *poldip2del* male flies, we next asked why older *poldip2del* male flies were completely infertile. Serendipitously, we found that nuclear DNA (nuDNA) was markedly fragmented in the mature sperm of 2-week-old *poldip2del* flies, but not in young *poldip2del* flies or wild-type flies at either age (Fig. 5A,B). Importantly, the ectopic expression of mitoExoIII, which can degrade mtDNA, suppressed the nuDNA fragmentation in mature sperm of old *poldip2del* flies (Fig. 5B). These results suggest a potential link between persistent mtDNA and nuDNA damage, which may contribute to the complete sterility of 2-week-old *poldip2del* flies.

During *Drosophila* spermatogenesis, nuDNA breaks occur to facilitate the histone-to-protamine transition from the post-meiotic

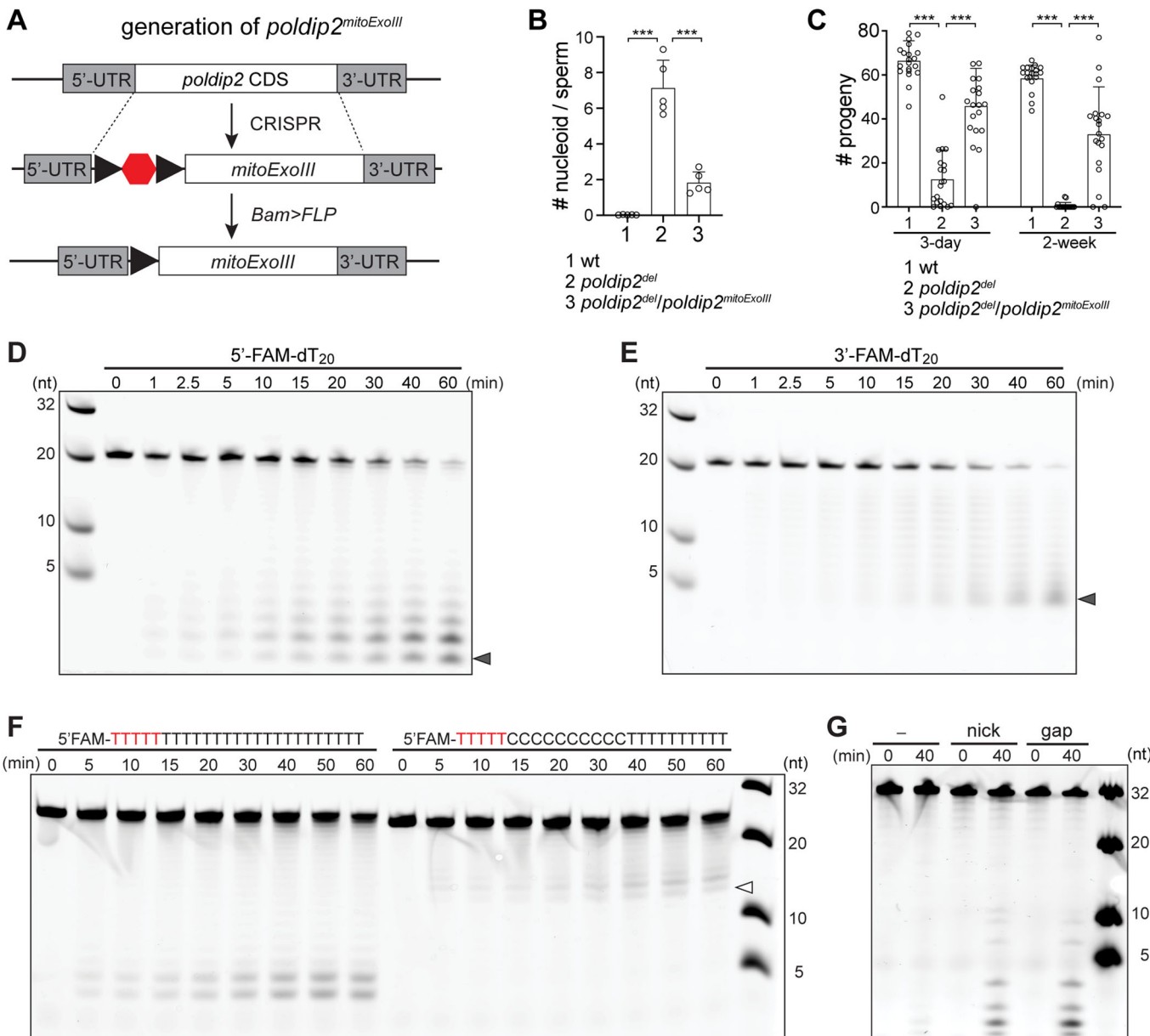

stage onwards. These breaks are repaired in the late elongation and sperm individualization stages. The timing of appearance and disappearance of nuDNA breaks was normal in *poldip2^del* flies (Fig. EV5A), suggesting that the nuDNA fragmentation is not caused by defects in chromatin remodeling. During apoptosis, EndoG is released from mitochondria and translocates to the nucleus, where it cleaves nuclear DNA, leading to nuclear genome fragmentation (Li et al, 2001). However, the *poldip2^del* and *endoG* double mutant fly, which had high levels of persistent mtDNA (Fig. 4H), showed clear nuDNA fragmentation (Fig. EV5D). These results indicate that sperm nuDNA fragmentation is independent of EndoG, and further substantiate a role of persistent mtDNA in nuclear genome fragmentation.

To investigate how persistent mtDNA is related to nuDNA fragmentation, we stained the testes with tetramethylrhodamine

methyl ester (TMRM) and MitoTracker Green, fluorescent dyes sensitive to mitochondrial membrane potential and mitochondrial mass, respectively. The ratio of TMRM intensity to that of MitoTracker Green, an indication of mitochondrial membrane potential, was decreased in mature sperm of young *poldip2^del* flies compared to the wt flies (Fig. EV5B). This reduced membrane potential was more pronounced in 2-week-old *poldip2^del* flies (Fig. EV5C). Additionally, reactive oxygen species (ROS) were also markedly increased in mature sperm of 2-week-old *poldip2^del* and double mutant flies (Figs. 5C,D and EV5E). Notably, this phenotype was rescued by expressing mitoExoIII in spermatids (Fig. 5C,D). These findings suggest that residual mtDNA may cause impaired mitochondrial respiration and increased oxidative stress over time, which could contribute to nuclear genome damage.

    

**Figure 3.  Poldip2 is a mitochondrial DNA exonuclease.**

(A) Genetic scheme of replacing the *poldip2* coding sequence (CDS) with a mitochondrially targeted *E. coli* Exonuclease III (mitoExoIII) in developing spermatids. The SV40 transcription termination sequence (hexagon) flanked by two FRT sites (arrowheads) allows conditional expression of mitoExoIII induced by Flippase (FLP). (B) Expression of mitoExoIII in spermatids reduced mitochondrial nucleoid numbers in mature sperm of 3-day-old *poldip2*$^{del}$ flies. (1) wt: *UAS-FLP/+; Bam-gal4, poldip2* $^{del}$ */+.* (2) *poldip2* $^{del}$: *+/+; Bam-gal4, poldip2* $^{del}$/*poldip2*$^{mitoExoIII}$; (3) *poldip2* $^{del}$/*poldip2*$^{mitoExoIII}$: *UAS-FLP/+; Bam-gal4, poldip2* $^{del}$/ *poldip2*$^{mitoExoIII}$. Each data point represents quantification from one seminal vesicle ($n = 5$). The data represent the mean± SD. Statistical analysis was performed using an unpaired *t* test. *P* values from left to right: \*\*\**P* = 3.2 × 10$^{-6}$, *P* = 4.8 × 10$^{-5}$. (C) Expression of mitoExoIII in spermatids rescued the fertility of both young and 2-week-old male *poldip2*$^{del}$ flies. The number of progeny per day is shown. (1) wt: *UAS-FLP/+; Bam-gal4, poldip2* $^{del}$ */+.* (2) *poldip2* $^{del}$: *+/+; Bam-gal4, poldip2* $^{del}$/*poldip2*$^{mitoExoIII}$; (3) *poldip2* $^{del}$/*poldip2*$^{mitoExoIII}$: *UAS-FLP/+; Bam-gal4, poldip2* $^{del}$/ *poldip2*$^{mitoExoIII}$. Each data point represents a biological replicate ($n = 20$). The data represent the mean± SD. Statistical analysis was performed using an unpaired *t* test. *P* values from left to right: \*\*\**P* = 2.6 × 10$^{-18}$, *P* = 3.2 × 10$^{-8}$, *P* = 3.0 × 10$^{-34}$, *P* = 4.9 × 10$^{-9}$. (D) Poldip2 exhibits 3′–5′ exonuclease activity. A 5′-6-FAM-labeled 20-nt poly(dT) (100 nM) was incubated with the Poldip2 protein (200 nM) at 37 °C and analyzed at the indicated time points. A ladder-like pattern of oligonucleotides ranging from monomer (arrowhead) to 19-mer was generated. (E) Poldip2 displays 5′–3′ exonuclease activity. A 3′-6-FAM-labeled 20-nt poly(dT) (100 nM) was incubated with the Poldip2 protein (200 nM) at 37 °C and analyzed at the indicated time points. The resulting products showed a ladder pattern ranging from 3-mer (arrowhead) to 19-mer. (F) Poldip2 degrades dC less efficiently. The 5′-6-FAM-labeled 25-nt poly(dT) or 25-nt poly(dTdC) (100 nM) was incubated with the Poldip2 protein (400 nM) at 37 °C and analyzed at the indicated time points. The 5′-end five nucleotides were protected from degradation by incorporating the internucleotide phosphorothioate bonds (red). Note the smallest degradation product is 15-mer (arrowhead) in the reaction of poly(dTdC), suggesting the stretch of dC inhibits the progression of the exonuclease. (G) Poldip2 degrades double-stranded DNA (dsDNA) with breaks. Three types of dsDNA, including blunt-ended dsDNA (−), dsDNA with a single nick (nick), and dsDNA with a single gap (gap), were generated using a 5′-6-FAM-labeled, 32-nt long oligonucleotide. The dsDNA substrates (100 nM) were incubated with the Poldip2 protein (1200 nM) at 37 °C and analyzed after 40 min. The molecular markers in this figure are an equal molar mixture of 5′-6-FAM-labeled 32-nt, 20-nt, 10-nt and 5-nt oligonucleotides and were loaded at a concentration of 50 nM for each. See also Table EV3. Source data are available online for this figure.

# Discussion

Previous studies show that EndoG is involved in degrading mtDNA during *Drosophila* spermatogenesis. However, EndoG only nicks dsDNA at $(dG)_n/(dC)_n$ tracks, whereas the metazoan mitochondrial genome possesses an unusually biased A/T composition (Gammage and Frezza, 2019; Kurabayashi and Ueshima, 2000; Wolstenholme, 1992; Chen et al, 2019), suggesting that other nucleases must be involved. Here, we discover that Poldip2 is a mitochondrial exonuclease, highly expressed in *Drosophila* testes, and essential for mtDNA removal in elongated spermatids. Poldip2 degrades DNA in both directions and prefers dA/dT nucleotides. The predicted structure of *Drosophila* Poldip2 differs from most common nuclease structures (Yang, 2011), which may account for its distinct nuclease properties.

The *poldip2* and *endoG* double mutant exhibited a stronger phenotype than either mutant individually, suggesting that these two enzymes may work in parallel in mtDNA removal. Alternatively, EndoG and Poldip2 may work in tandem, with EndoG generating initial breaks on mtDNA, followed by Poldip2 degrading the nicked DNA. The removal of mtDNA is delayed but eventually executed in *endoG* mutant flies, which exhibit normal fertility (DeLuca and O'Farrell, 2012), suggesting that other mechanisms independent of EndoG may generate breaks in mtDNA, facilitating its degradation by Poldip2 (Fig. 5E).

Developmental declines in mtDNA content have also been observed during sperm development in mammals (Lee et al, 2023; Luo et al, 2013; May-Panloup et al, 2003; Larsson et al, 1997), albeit to varying extents. The human Poldip2 ortholog (hPoldip2) can rescue the male fertility defect of *poldip2*$^{del}$ flies (Fig. 1E), suggesting that hPoldip2 may also function as a mitochondrial exonuclease. Unlike the enriched expression of Poldip2 in *Drosophila* testes, hPoldip2 is ubiquitously expressed across various tissues (Fagerberg et al, 2014), indicating a potentially broader role for hPoldip2 in maintaining mtDNA homeostasis across diverse cell types. Mitochondrial genomes have a substantially higher mutation rate than nuclear genomes in metazoans (Lynch et al, 2006). However, the repertoire of repair pathways for mtDNA is limited (Fu et al, 2020). Studies in

mammalian cells have demonstrated that damaged mtDNA molecules are rapidly degraded (Srivastava and Moraes, 2001; Nissanka et al, 2018), followed by repopulation with intact genomes. Given the presence of multiple copies of the mitochondrial genome in each cell and the relatively higher cost of repair mechanisms (Fu et al, 2020), degradation of damaged mtDNA might be an effective way to preserve fidelity. It would be intriguing to investigate the potential roles of Poldip2 in regulating mtDNA homeostasis and maintaining mtDNA integrity.

In multicellular organisms, the egg is furnished with maternally derived organelles and macromolecules, including RNAs and proteins deposited with defined polarity and spatial patterns (Kimelman and Martin, 2012) to support the rapid early embryonic cycles and instruct the subsequent pattern formation. In contrast, the mature sperm is a "stripped down" cell. All macromolecules and organelles in the cytoplasm, including ribosomes, ER, and Golgi, except for mitochondria and the axoneme, are cleared (Fabian and Brill, 2012; Barratt et al, 2009). This clearance process not only prevents the deposition of sperm-derived proteins and mRNAs that could disrupt early embryonic cycles and patterning but also streamlines sperm shape for effective movement (Kimelman and Martin, 2012; Barratt et al, 2009). It has been demonstrated that mtDNA replisomes are enriched in two membrane-spanning structures and tethered to ER-mitochondrial contacts (Meeusen and Nunnari, 2003; Lewis et al, 2016), which are frequently observed in developing spermatids (Fabian and Brill, 2012; Tokuyasu, 1974). If not degraded, mtDNA could be associated with ICs indirectly through these structures in individualizing spermatids, impeding the progression of traveling ICs. This proposition explains the association of nucleoids with ICs in the *poldip2* and *endoG* double mutant flies and the individualization defects in these flies (Fig. 4G). However, we cannot rule out the possibility that the compromised mitochondrial activity may also contribute to the individualization defects.

Persistent mtDNA could potentially cause mito-nuclear imbalance. In mature sperm, nuclear genes' expression is completely shut down. Persistent mtDNA could produce excessive, unassembled mtDNA-encoded electron transport chain subunits, which may

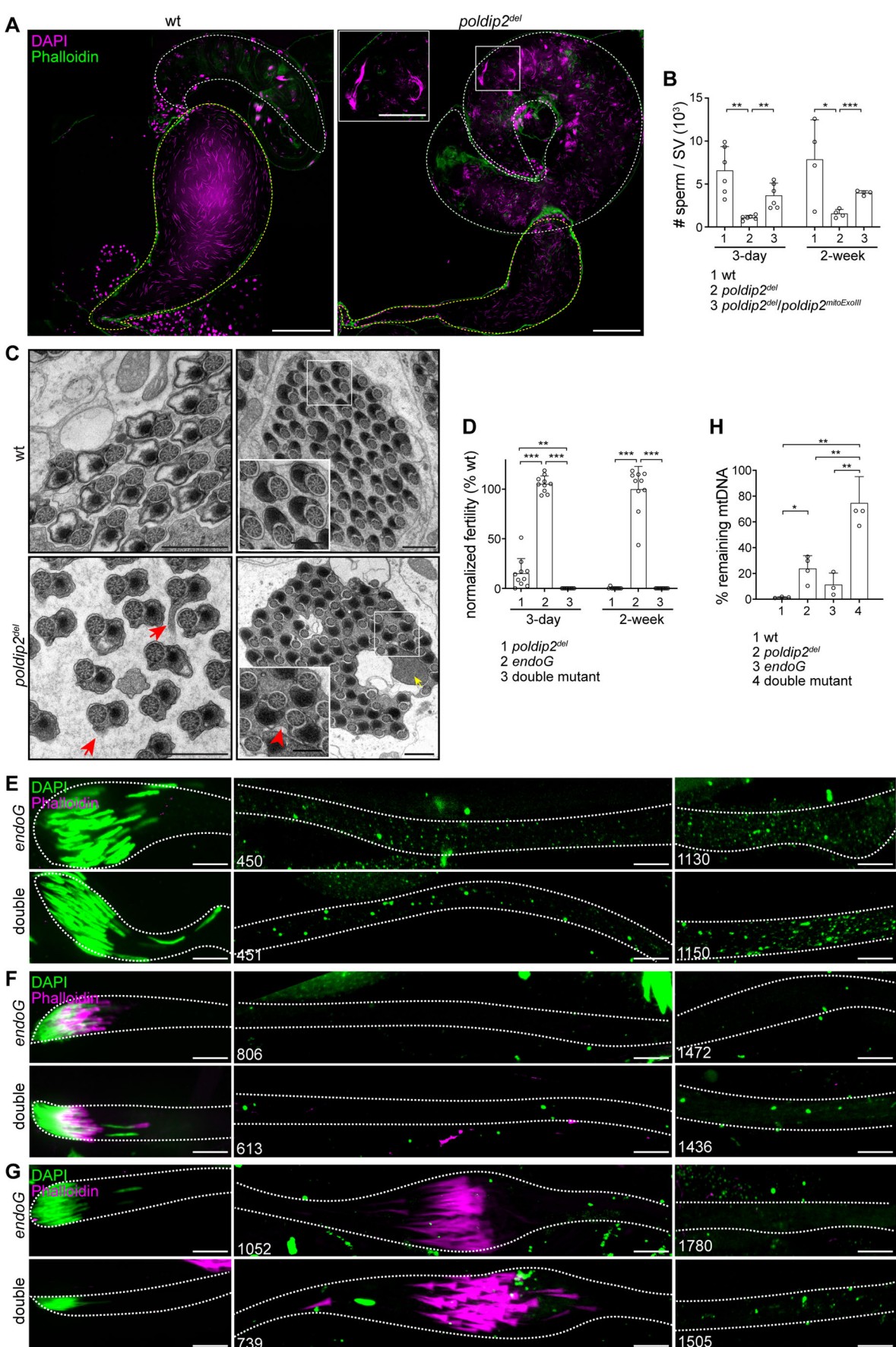

◀

**Figure 4. Persistent mtDNA impedes spermatid individualization.**

(A) Representative images showing the coiling region (white dashed line) and seminal vesicle (yellow dashed line) of $w^{1118}$ (wt) and $poldip2^{del}$ testes. Bar, 100 μm. Inset, the coiling region of $poldip2^{del}$ testes accumulates many needle-shaped nuclei stained with DAPI (magenta), some of which remain bundled together. Phalloidin: green. Bar, 50 μm. (B) Fewer mature sperm are present in the seminal vesicles of both young and 2-week-old $poldip2^{del}$ flies compared with the wild-type control, a deficiency that can be rescued by expressing mitoExoIII in spermatids. The number of mature sperm nuclei was quantified in each seminal vesicle of the indicated genotypes. (1) wt: *UAS-FLP/* +; *Bam-gal4, poldip2* $^{del}$ /+. (2) $poldip2$ $^{del}$: +/+; *Bam-gal4, poldip2* $^{del}$/$poldip2^{mitoExoIII}$; (3) $poldip2$ $^{del}$/$poldip2^{mitoExoIII}$: *UAS-FLP/+; Bam-gal4, poldip2* $^{del}$/ $poldip2^{mitoExoIII}$. Each data point represents quantification from one seminal vesicle ($n = 6, 4$). The data represent the mean± SD. Statistical analysis was performed using an unpaired $t$ test. $P$ values from left to right: **$P = 0.00068$, **$P = 0014$, *$P = 0.035$, ***$P = 0.0001$. (C) Representative transmission electron microscopy (TEM) images of cross sections of *Drosophila* testes from 3-day-old flies showing individualized spermatid cysts. In $poldip2^{del}$ testes, red arrows denote spermatids with incomplete membrane contour; red arrowheads denote two connected spermatids; yellow arrow denotes abnormal mitochondrial derivate structures. Bar, 1 μm. Bar in insets, 500 nm. (D) The double mutant flies are completely sterile, in contrast to the normal fertility of the endoG mutant. The fertility of male $poldip2^{del}$, $endoG$ ($endoG^{MB07150/KO}$) and double mutant ($endoG^{MB07150/KO}$; $poldip2^{del}$) was normalized to that of $w^{1118}$ male flies and plotted ($n = 10$). The data represent the mean ± SD. Statistical analysis was performed using an unpaired $t$ test. $P$ values from left to right: **$P = 0.005$, ***$P = 2.1 \times 10^{-12}$, ***$P = 4.0 \times 10^{-19}$, ***$P = 6.6 \times 10^{-11}$, ***$P = 6.2 \times 10^{-11}$. (E–G) Representative images showing the isolated spermatid bundles of the elongating (E), fully elongated (F), and individualization (G) stages stained for DNA (DAPI) and actin cones (Phalloidin) in $endoG$ ($endoG^{MB07150/KO}$) and double mutant ($endoG^{MB07150/KO}$; $poldip2^{del}$). DAPI stains both nuDNA and mtDNA in isolated spermatid bundles. Note the disorganized actin cone structures in the double mutant. Dashed lines mark bundles' boundary. Numbers indicate the distance (μm) from the anterior tip of the spermatid. Bar, 10 μm. (H) Quantification of remaining mitochondrial nucleoids in fully elongated spermatids of $w^{1118}$ (wt), $poldip2^{del}$, $endoG$ ($endoG^{MB07150/KO}$) and double mutant ($endoG^{MB07150/KO}$; $poldip2^{del}$) flies. Total mitochondrial nucleoids measured in volumes per spermatid in the fully elongated stage were normalized to that of the elongating stage in each genotype. Each data point represents a spermatid ($n = 3, 4$). The data represent the mean ± SD. Statistical analysis was performed using an unpaired $t$ test. $P$ values from left to right: *$P = 0.013$, **$P = 0.0018$, **$P = 0.0042$, **$P = 0.0045$. Source data are available online for this figure.

impair mitochondrial respiration and lead to the generation of damaging free radicals (Zhao et al, 2002; Sutandy et al, 2023). Supporting this idea, mature sperm of old $poldip2^{del}$ flies exhibited increased ROS levels (Fig. 5C,D), along with markedly fragmented nuclear genomes (Fig. 5A,B). A study in mice also showed that sperm mobility is negatively correlated with mtDNA copy numbers (Luo et al, 2013). Therefore, mtDNA removal appears essential for two key aspects of male reproductive biology: the effective removal of cytoplasm during sperm development and the prevention of potential mito-nuclear imbalance in mature sperm. The stringent uniparental inheritance of the mitochondrial genome, one of the most mysterious genetic phenomena in multicellular organisms, might be a prerequisite for the asymmetry between two gametes in sexual reproduction.

# Methods

### Reagents and tools table

| Reagent/resource | Reference or source | Identifier or catalog number |
|---|---|---|
| **Experimental models** | | |
| *Drosophila* S2 cell | Drosophila Genomics Resource Center | S2-DGRC |
| $w^{1118}$ | Bloomington Drosophila Stock Center | 5905 |
| $EndoG^{MB07150}$ | Bloomington Drosophila Stock Center | 26072 |
| UAS-FLP | Bloomington Drosophila Stock Center | 62139 |
| Vasa-Cas9 | Bloomington Drosophila Stock Center | 51323 |
| *Bam-gal4* | Chen and McKearin, 2003 | N/A |
| *Nanos-gal4* | Bloomington Drosophila Stock Center | 4937 |
| *MTD-gal4* | Bloomington Drosophila Stock Center | 31777 |
| *ATG7 RNAi* | Bloomington Drosophila Stock Center | 34369 |

| Reagent/resource | Reference or source | Identifier or catalog number |
|---|---|---|
| *hs-FLP; Act > CD2 > GAL4, UAS-mCD8::mCherry* | Zhang et al, 2024 | N/A |
| $w^{1118}$ ($mt:ND2^{del1}$) | Xu, 2008 | N/A |
| TFAM-mNeonGreen | Zhang et al, 2024 | N/A |
| DJ-MTS-DsRed | Politi et al, 2014 | N/A |
| UASz-poldip2 | This study | N/A |
| $poldip2^{del}$ | This study | N/A |
| $endoG^{KO}$ | This study | N/A |
| Poldip2-mNeonGreen | This study | N/A |
| $poldip2^{mitoExoIII}$ | This study | N/A |
| P{$poldip2$} mini gene | This study | N/A |
| P{h$Poldip2$} mini gene | This study | N/A |
| **Recombinant DNA** | | |
| pAW | Drosophila Genomics Resource Center | 1127 |
| pattB | Drosophila Genomics Resource Center | 1420 |
| poldip2 ORF clone | Drosophila Genomics Resource Center | 3372 |
| hPoldip2 ORF clone | GeneCopoeia | L0059 |
| pIB-V5/His | Thermo Scientific | V802001 |
| pENTR3C | Thermo Scientific | A10464 |
| pET21b | Novagen | 69741 |
| pU6-BbsI-chiRNA | Addgene | 45946 |
| pUASz 1.0 | Drosophila Genomics Resource Center | 1431 |
| pOT2 | Berkeley Drosophila Genome Project | https://www.fruitfly.org/about/methods/pOT2vector.html |
| FRT-SV40 PolyA-FRT /pBluescript KS (-) | Tran et al, 2012 | N/A |
| SOD-cGFP | Zhang et al, 2015 | N/A |
| pIB-V5-Poldip2-mCherry-nGFP | This study | N/A |
| pAW-Poldip2-mcherry | This study | N/A |
| pET21b-Poldip2 | This study | N/A |

| Reagent/resource | Reference or source | Identifier or catalog number |
|---|---|---|
| **Antibodies** | | |
| Anti-Poldip2 | This study | N/A |
| Anti-Polyglycylated-tubulin | Sigma-Aldrich | MABS276 |
| Anti-BrdU | abcam | ab152095 |
| Alexa Fluor 647 Phalloidin | Invitrogen | A22287 |
| Alexa Fluor 568 goat anti-mouse IgG | Invitrogen | A-11004 |
| Alexa Fluor 568 goat anti-rabbit IgG | Invitrogen | A-11011 |
| **Oligonucleotides and other sequence-based reagents** | | |
| PCR primers | This study | Table EV1 |
| ddPCR primers/probe | This study | Table EV2 |
| Oligos for nuclease assay | This study | Table EV3 |
| Oligos for mtDNA smFISH | Hurd et al, 2016 | N/A |
| **Chemicals, enzymes, and other reagents** | | |
| Gateway LR-clonase II | Thermo Scientific | 11791020 |
| In-Fusion Snap Assembly | Takara | 638955 |
| Paraformaldehyde | Electron Microscopy Sciences | 15710 |
| PBS, pH 7.4 | Quality Biological | 114-058-101 |
| Triton X-100 | Sigma | T9284 |
| Tris, pH 6.8 | Quality Biological | 351-091-101 |
| Tris, pH 7.5 | Quality Biological | 351-048-101 |
| Tris, pH 8.0 | Quality Biological | 351-007-101 |
| KCl | KD Medical | CAC-5320 |
| NaCl | KD Medical | RGE-3270 |
| EDTA | KD Medical | RGC-3130 |
| $MgCl_2$ | Quality Biological | 340-034-721 |
| $CaCl_2$ | Quality Biological | 351-130-721 |
| Sodium phosphate dibasic | Quality Biological | RGF-3282 |
| Sodium phosphate monobasic | Quality Biological | RGF-3280 |
| BSA | Sigma | A8806 |
| Vectashield mounting medium with DAPI | Vector Laboratories | H-1500 |
| Schneider's Drosophila medium | Gibco | 21720 |
| FBS | Gibco | A5670701 |
| Penicillin–streptomycin | Gibco | 15140122 |
| Concanavalin A | Sigma | C0412 |
| Effectene transfection reagent | Qiagen | 301425 |
| Picogreen | Invitrogen | P11496 |
| Lab-Tek II chambered coverglass | Nunc | 155409 |
| siliconized coverslip | Hampton Research | HR3-215 |
| poly-L-lysine | Sigma | P4707 |
| DAPI | Sigma | D9542 |
| QIAamp DNA Micro kit | Qiagen | 56304 |
| EcoRI | New England Biolabs | R3101 |
| ddPCR Supermix for Probes | Bio-Rad | 186-3023 |

| Reagent/resource | Reference or source | Identifier or catalog number |
|---|---|---|
| BL21(DE3) competent cells | Thermo Scientific | EC0114 |
| IPTG | Sigma | I6758 |
| Imidazole | Sigma | I2399 |
| Glycerol | Invitrogen | 15514011 |
| β-mercaptoethanol | Gibco | 21985023 |
| DTT | Sigma | D0632 |
| Lysozyme | Sigma | L6876 |
| EDTA-free protease inhibitor | Roche | 11873580001 |
| HisTrap | Cytiva | 17524802 |
| HiTrap heparin | Cytiva | 17040703 |
| Superdex 200 increase 10/300 GL | Cytiva | 28990944 |
| Bradford plus protein assay reagents | Thermo Scientific | 23238 |
| Coomassie InstantBlue Protein Stain | Abcam | ab119211 |
| Nuclease-free duplex buffer | IDT | 11-01-03-01 |
| TBE Urea sample buffer | Thermo Scientific | LC6876 |
| Denaturing polyacrylamide gel (SequaGel) | National Diagnostics | EC-833 |
| Glutaraldehyde | Electron Microscopy Sciences | 16020 |
| Sodium cacodylate buffer | Sigma | 97068 |
| Osmium tetroxide | Sigma | 19100 |
| Potassium ferrocyanide | Sigma | 60279 |
| Uranyl acetate | Electron Microscopy Sciences | 22400 |
| Thiocarbohydrazide | Electron Microscopy Sciences | 21900 |
| Lead Nitrate | Electron Microscopy Sciences | 17900 |
| Epon-Araldite | Electron Microscopy Sciences | 13940 |
| APO-BrdU kit | Phoenix Flow Systems | AU1001 |
| Sucrose | Sigma | 84097 |
| Potassium acetate | Sigma | P1190 |
| Sodium acetate | Millipore | 567422 |
| EGTA | KD Medical | PKE-0460 |
| SSC buffer | Corning | 46-020-CM |
| Tween-20 | Promega | H5152 |
| Formamide | Sigma | 47671 |
| Dextran sulfate | Sigma | S4030 |
| Vanadyl ribonucleoside complex | Sigma | 94742 |
| Tetramethylrhodamine (TMRM) | Invitrogen | I34361 |
| MitoTracker Green | Invitrogen | M7514 |
| CellROX Deep Red | Thermo Scientific | C10422 |
| Heptane | Sigma | 246654 |
| **Software** | | |
| GraphPad Prism 10.0 | https://www.graphpad.com/ | N/A |

 

| Reagent/resource | Reference or source | Identifier or catalog number |
|---|---|---|
| Fiji 2.14.0 | https://fiji.sc; https://doi.org/10.1186/s12859-017-1934-z | N/A |
| QuantaSoft Analysis Pro | Bio-Rad | N/A |
| **Other** | | |
| Fly injection | Bestgene | N/A |
| Leica SP8 confocal system | Leica | N/A |
| PerkinElmer UltraView confocal | PerkinElmer | N/A |
| Visitech instant structured illumination microscope (iSIM) | BioVision | N/A |
| Nikon CSU-W1 SoRa confocal system | Nikon | N/A |
| QX200 ddPCR system | Bio-Rad | N/A |
| ÄKTA pure protein purification system | Cytiva | N/A |
| Typhoon biomolecular imager | GE | N/A |
| Trainable Weka Segmentation | https://doi.org/10.1093/bioinformatics/btx180 | N/A |

## Fly stocks and husbandry

*Drosophila melanogaster* was reared in a humidity-controlled incubator under a 12-hour light-dark cycle at 25 °C using standard cornmeal molasses agar media. The following fly stocks were used: *w^{1118}*, *EndoG^{MB07150}* (Bloomington Drosophila Stock Center, BDSC, stock 26072), *UAS-FLP* (BDSC, stock 62139), *Nanos-gal4* (BDSC, stock 4937), *Bam-gal4* (Chen and McKearin, 2003), *MTD-gal4* (BDSC, stock 31777), ATG7 RNAi (BDSC, stock 34369), *w^{1118}* (*mt:ND2^{del1}*) (Xu, 2008), TFAM-mNeonGreen (Zhang et al, 2024), *hs-FLP; Act > CD2 > GAL4, UAS-mCD8::mCherry* (Zhang et al, 2024), and DJ-MTS-DsRed (Politi et al, 2014). The *w^{1118}* strain was used as the control fly unless otherwise specified.

## Molecular cloning and transgenic flies

To construct Poldip2-mCherry plasmid that expresses in *Drosophila* S2 cell culture, the coding sequence of *poldip2* was PCR amplified from the DGRC (Drosophila Genomics Resource Center) Gold collection (DGRC, stock 3372), fused with C-terminal mCherry tags, and cloned into the pENTR3C vector. Subsequently, the plasmids were recombined into the pAW vector (DGRC, stock 1127) using Gateway LR-clonase II (Thermo Scientific, 11791020). For the Poldip2-mCherry-nGFP plasmid, the *poldip2* coding sequence was fused with a C-terminal mCherry tag followed by N-terminal half of GFP sequence and inserted into the pIB-V5 vector. Plasmid SOD2-cGFP was previously published (Zhang et al, 2015). Poldip2-6His construct was generated by cloning the *poldip2* coding sequence into the pET21b vector, fused with a C-terminal 6His tag.

To generate *poldip2* mini gene, ~5 kb of 5′ and 3′ flanking sequences of the *poldip2* gene were PCR amplified from *w^{1118}* genomic DNA and cloned into the pattB vector (DGRC, stock 1420) using the In-Fusion cloning method (Takara, 638955). The resulting construct was designated as the pattB-poldip2-mini. For the *hPoldip2* genomic rescue transgene construct, the coding region

of *poldip2* was substituted with that of *hPoldip2* (GeneCopoeia) on the pattB-poldip2-mini plasmid. To generate UASz-Poldip2-mcherry transgene, the *poldip2* coding sequence fused with C-terminal mCherry tag was cloned into the pUASz1.0 vector (DGRC, stock 1431). Subsequently, the transgenic flies were produced by injecting the plasmids into 25C6 at the attP40 site via PhiC31 integrase-mediated transgenesis (BestGene).

## Generation of CRISPR knock-out and knock-in flies

The knock-out and knock-in lines, including *poldip2^{del}*, *endoG^{KO}*, Poldip2-mNeonGreen and *poldip2^{mitoExoIII}*, were generated using standard CRISPR methods (Gratz et al, 2015). For knock-out lines *poldip2^{del}* and *endoG^{KO}*, the non-homologous end joining (NHEJ) approach was employed, involving the design of two guide RNAs (gRNAs) to delete most of the coding region of the respective genes. The guide RNA recognition sites for *poldip2* deletion were GCGATGTCCCATGCTCCACAGGG (Table EV1, gRNA1) and GCGCGACGATAGCGATTAAGAGG (Table EV1, gRNA2), while those for *endoG* deletion were GGCAGCCGAAACAGTTCCAAAGG (Table EV1, gRNA3) and GAGAGCGTGGAACGCTCGGCGGG (Table EV1, gRNA4). The synthesized gRNA sequences were annealed and cloned into the pU6-BbsI-chiRNA vector (Addgene #45946). Subsequently, the two plasmids carrying the gRNAs for each gene were injected into Vasa-Cas9 (BDSC, stock 51323) expressing embryos (BestGene). Eclosed flies were crossed with *w^{1118}* flies, and the progeny carrying the deletions were screened using PCR. Sanger sequencing was performed to verify the deleted sequence.

For Poldip2-mNeonGreen knock-in line, *mNeonGreen* coding sequence was inserted into the C-terminus of *poldip2* gene at the genomic locus via the homolog-directed repair mechanism of CRISPR. A homology donor containing 1000 base pairs upstream of the stop codon of *poldip2* (left homology arm), a linker sequence (Waldo et al, 1999), followed by the *mNeonGreen* coding sequence, and 1000 base pairs downstream of the stop codon of *poldip2* (right homology arm), was cloned into the pOT2 vector. The resulting donor plasmid was co-injected with the guide RNA plasmid (Table EV1, gRNA2) into Vasa-Cas9 (BDSC, stock 51323) expressing embryos (BestGene). Eclosed flies were crossed with *w^{1118}* flies, and the progeny carrying *mNeonGreen* insertions were screened via PCR. The sequences of the *poldip2* genomic region and *mNeonGreen* were verified using Sanger sequencing. Poldip2-mNeonGreen is functional, as homozygous *mNeonGreen* knock-in flies were healthy.

The *poldip2^{mitoExoIII}* line was generated by replacing the coding region of *poldip2* gene with the FRT-SV40-FRT-mitoExoIII cassette through CRISPR. First, a homology donor spanning 1000 base pairs upstream of the start codon (left homology arm) and the 1000 base pairs downstream of the stop codon (right homology arm) of the *poldip2* gene was PCR amplified and cloned into the pOT2 vector. Concurrently, the FRT-SV40 PolyA-FRT /pBluescript KS (−) plasmid (Tran et al, 2012) was utilized, with a citrate synthase mitochondrial-targeting sequence (MTS, mito) followed by the *E. coli xthA* gene (coding Exonuclease III) cloned after the second FRT site via In-Fusion cloning (Takara). Then, the entire FRT-SV40-FRT-mitoExoIII sequence was recovered using NotI and EcoRI digestion, replacing the *poldip2* coding region on the homology donor plasmid via In-Fusion cloning. The resulting

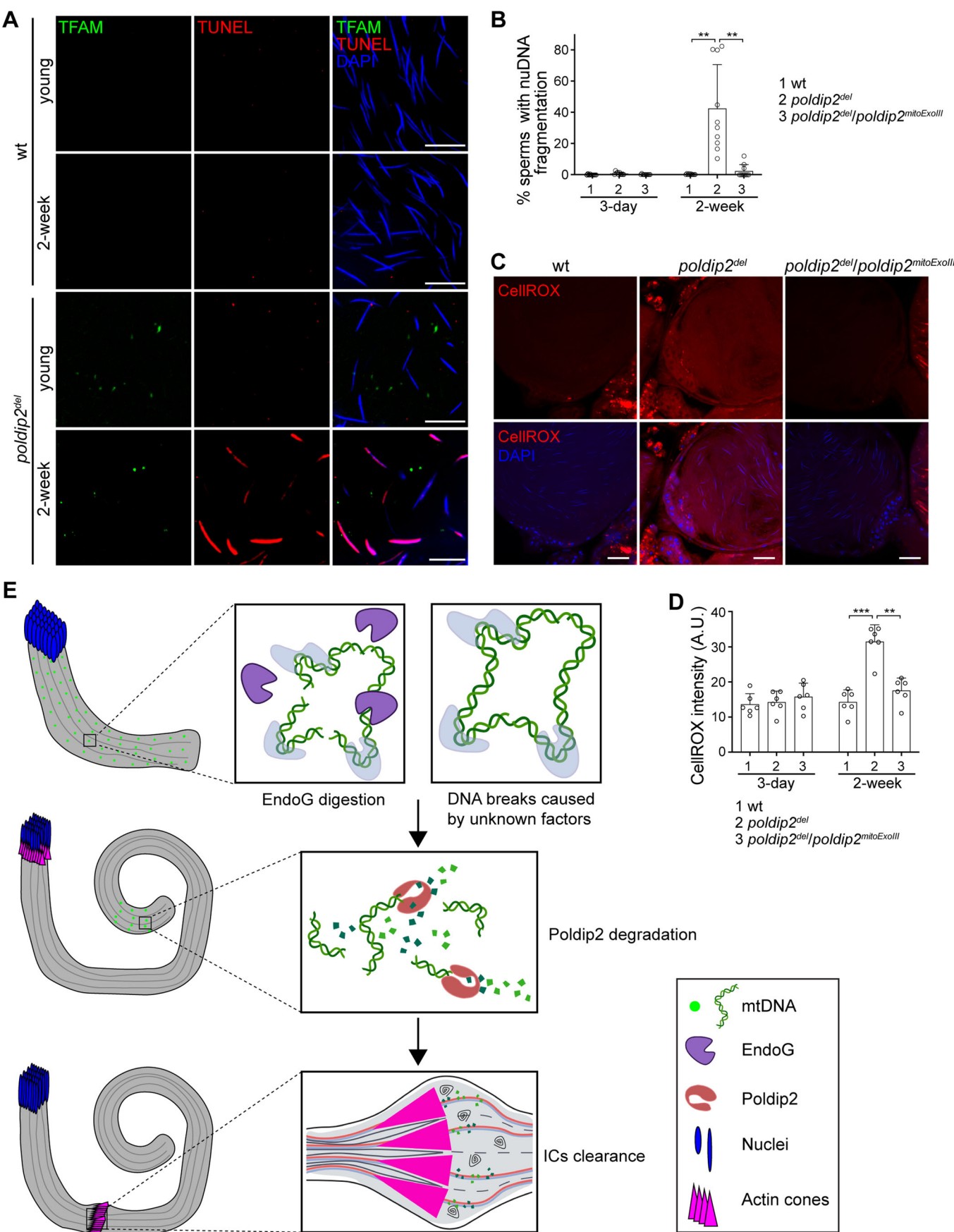

◀ **Figure 5. Persistent mtDNA in mature sperm damages the nuclear genome.**

(A) Representative images of a TUNEL assay showing nuDNA breaks/fragmentation in 2-week-old *poldip2*$^{del}$ mature sperm. TFAM-mNeonGreen labels mitochondrial nucleoids (green), and DAPI (blue) stains nuDNA of mature sperm in seminal vesicles. Red: TUNEL signal. Bar, 10 μm. (B) Expression of mitoExoIII in spermatids reduces nuDNA breaks/fragmentation in *poldip2*$^{del}$ mature sperm. Quantification was performed by normalizing the nuDNA breaks/fragmentation stained with the TUNEL assay to the total sperm numbers stained by DAPI in a seminal vesicle. (1) wt: *UAS-FLP*/+; *Bam-gal4, poldip2* $^{del}$ /+. (2) *poldip2* $^{del}$: +/+; *Bam-gal4, poldip2* $^{del}$/*poldip2*$^{mitoExoIII}$; (3) *poldip2* $^{del}$/*poldip2*$^{mitoExoIII}$: *UAS-FLP*/+; *Bam-gal4, poldip2* $^{del}$/ *poldip2*$^{mitoExoIII}$. Each data point represents quantification from one seminal vesicle ($n = 10$). The data represent the mean± SD. Statistical analysis was performed using an unpaired *t* test. *P* values from left to right: \*\*$P = 0.00017$, $P = 0.00031$. (C) Representative images of CellROX Deep Red staining of seminal vesicles from 2-week-old wt, *poldip2*$^{del}$ and *poldip2*$^{del}$ /*poldip2*$^{mitoExoIII}$ flies. Red: CellROX Deep Red; Blue: nuDNA stained by DAPI. Bar, 20 μm. (D) Quantification of CellROX intensity, as the measure of ROS levels in both young and 2-week-old flies with indicated genotypes. (1) wt: *UAS-FLP*/+; *Bam-gal4, poldip2* $^{del}$ /+. (2) *poldip2* $^{del}$: +/+; *Bam-gal4, poldip2* $^{del}$/*poldip2*$^{mitoExoIII}$; (3) *poldip2* $^{del}$/*poldip2*$^{mitoExoIII}$: *UAS-FLP*/+; *Bam-gal4, poldip2* $^{del}$/ *poldip2*$^{mitoExoIII}$. A.U., arbitrary unit. Each data point represents quantification from one seminal vesicle ($n = 6$). The data represent the mean± SD. Statistical analysis was performed using an unpaired *t* test. \*\*\*$P = 3.6 \times 10^{-5}$, \*\*$P = 0.0002$. (E) Proposed model of pre-fertilization mtDNA removal. In elongating spermatids, mitochondria undergo dramatic structural changes, potentially sensitizing mitochondrial nucleoids. This may trigger the EndoG-dependent mtDNA nicking or DNA breaks through other unknown mechanisms, initiating the clearance of mtDNA. In the final stage of spermatid elongation, the abrupt expression of Poldip2 leads to the complete degradation of mtDNA. During individualization, the individualization complexes (ICs) progress down the spermatids, gathering any remaining oligonucleotides, and ultimately discarding them in waste bags. Consequently, mature sperm are devoid of mtDNA. Source data are available online for this figure.

donor plasmid was injected along with two guide RNA plasmids (Table EV1, gRNA1 and gRNA2), into Vasa-Cas9 (BDSC, stock 51323) embryos (BestGene). Eclosed flies were crossed with *w*$^{1118}$ flies, and the progeny carrying the FRT-SV40-FRT-mitoExoIII cassette insertions were screened via PCR. The sequence of *poldip2* genomic region and the insertions were verified using Sanger sequencing.

To ensure a homogeneous genetic background, all CRISPR lines were backcrossed for at least six generations into a *w*$^{1118}$ background.

## Male fertility assay

Individual 2-day-old males of the specified genotypes were paired with two virgin *w*$^{1118}$ females. Every 3 days, the male flies were transferred to fresh vials and paired with another two virgin *w*$^{1118}$ females. The adult progeny from each male fly were counted.

## Antibodies

Antibodies used in this study were as follows: A custom antibody against Poldip2 was generated in rabbits using recombinant full-length protein as the antigen (GenScript). Mouse polyglycylated tubulin antibody (1:1000, Sigma-Aldrich, MABS276); rabbit anti-BrdU (1:200, abcam, ab152095); Alexa Fluor 647 Phalloidin (1:50, Invitrogen, A22287); Alexa Fluor 568 goat anti-mouse IgG (1:200, Invitrogen, A-11004); and Alexa Fluor 568 goat anti-rabbit IgG (1:200, Invitrogen, A-11011).

## Immunostaining of *Drosophila* testes

*Drosophila* testes with the specified genotypes were dissected in Schneider's *Drosophila* medium supplemented with 10% fetal bovine serum (FBS) and fixed in PBS containing 4% paraformaldehyde (Electron Microscopy Sciences, 15710) for 20 min. After three washes with PBS, the tissues were permeabilized in 0.5% Triton X-100 in PBS for 20 min. Subsequently, the testes were incubated with blocking solution (PBS, 0.2% BSA, 0.1% Triton X-100) for 1 h before being incubated with primary antibodies diluted in blocking solution at 4 °C overnight. Following three washes with blocking solution, the tissues were incubated with Alexa Fluor 647 Phalloidin and Alexa Fluor-conjugated secondary antibodies for 1 h

at room temperature. Finally, the testes were mounted in Vectashield mounting medium with DAPI (Vector Laboratories, H-1500). Images were acquired using a Leica SP8 confocal system (Leica HC PL APO 63×/1.4 oil lens; LAS X acquisition software version 3.5.7; scan speed 400 Hz; Pinhole 1 A.U; Excitation at 405, 488, 561, and 640 nm; z-stacks with 1 μm per step). A tile scan was performed to obtain stitched images of the whole testes and seminal vesicles. Image processing was performed using Fiji software (NIH).

## Cell culture

S2 cells were cultured in Schneider's *Drosophila* medium supplemented with 10% FBS and penicillin–streptomycin (100 U/mL) following standard procedures. One day before transfection, S2 cells were seeded onto eight-well glass-bottom chambered coverslips pre-treated with a 0.5 mg/ml Concanavalin A (Sigma, C0412) solution. Plasmids were transfected into S2 cells using Effectene transfection reagent (Qiagen, 301425). Fluorescent images were acquired using a PerkinElmer UltraView confocal system (Zeiss Plan-apochromat 63×/1.4 oil lens; Volocity acquisition software; Hamamatsu Digital Camera C10600 ORCA-R2) ~48 h after transfection.

To stain mitochondrial nucleoids in S2 cells, Picogreen reagent (Invitrogen, P11496) was diluted 1:300 in cell culture medium and incubated with the cells for 30 min (protected from light). After rinsing with PBS three times, fluorescent images were acquired.

## Staining of spermatid cysts and quantification of mitochondrial nucleoids

Isolation and staining of spermatid cysts were carried out following a previously published protocol with minor modifications (DeLuca and O'Farrell, 2012). Testes were dissected from 2- to 3-day-old male flies in ice-cold TB buffer (10 mM Tris pH 6.8, 183 mM KCl, 47 mM NaCl, 1 mM EDTA) and transferred to a small drop of TB (~10 μl) on a siliconized coverslip (Hampton Research, HR3-215). The base of the testis was gently torn off using forceps. While holding the unopened anterior tip of the testis with forceps, the contents were extruded using a glass capillary tube as a squeegee. The sample was then sandwiched between a poly-L-lysine (0.01%, Sigma)-treated slide and the coverslip. The sandwich was briefly frozen in liquid nitrogen for 20–30 s, the coverslip was removed

with a razor blade, and the slide containing the samples was incubated in ice-cold absolute ethanol. The tissues were fixed with 3.7% paraformaldehyde in PBS for 20 min, washed twice with PBS for 5 min each, and permeabilized with 0.1% Triton X-100 in PBS for 30 min. After washing in PBS twice, the samples were incubated with Alexa Fluor 647 Phalloidin diluted in blocking solution (PBS, 0.2% BSA, 0.1% Triton X-100) for 2 h at 37 °C in a humid chamber. Samples were washed three times with PBS, stained with DAPI (1 µg/ml in PBS, Sigma) for 30 min at room temperature, followed by washing in PBS three times and mounting in Vectashield mounting medium with DAPI. Images were acquired on a PerkinElmer UltraView confocal system (Zeiss Plan-apochromat 63×/1.4 oil lens; Volocity acquisition software; Hamamatsu Digital Camera C10600 ORCA-R2; z-stacks with 0.5 µm per step). A tile scan was performed to obtain stitched images of whole spermatid cysts.

To assess the number and size of mitochondrial nucleoids in spermatid bundles, image analysis was performed using Fiji (NIH). Initially, various regions of interest (ROIs) along the length of a spermatid bundle were selected and duplicated as z-stack images. The length of the spermatid bundle in each ROI, as well as the distance of the ROI from the nuclear head, was measured using "Measure" function. Subsequently, within each ROI, the spermatid cyst was outlined, and the "Clear Outside" function was applied to focus solely on the area within the spermatid cysts for analysis. Next, in the 405 nm (DAPI) channel, individual mitochondrial nucleoids were segmented using the Fiji plugin "Trainable Weka Segmentation 3D". The resulting hyperstack "probability maps" were further analyzed using the "3D Objects Counter" function. This enabled the quantification of the number of segmented mitochondrial nucleoids, as well as the volume of each individual nucleoid within the ROI. The nucleoid density was calculated by dividing the total number or total volume of mitochondrial nucleoids by the length of the spermatid bundle in the respective ROI. To determine the total number or volume of mitochondrial nucleoids in each spermatid bundle, nucleoid density at various points along the length of the bundle was plotted (Figs. 2E and EV2B) using GraphPad Prism. The area under the curve, which represents the total nucleoid numbers or volumes in 64 spermatids, was subsequently calculated.

## Quantification of paternal mtDNA copy number using droplet digital PCR (ddPCR)

To quantify mtDNA copy number in mature sperm, $w^{1118}$ ($mt:ND2^{del1}$) females were crossed with $w^{1118}$ ($mt:wt$) or $poldip2^{del}$ ($mt:wt$) males. For each cross, 30 2-day-old virgin females were mated with 30 2-day-old virgin males for 24 h at 25 °C. Spermathecae, the sperm storage organ, from mated or virgin control $w^{1118}$ ($mt:ND2^{del1}$) females were dissected in PBS. Tissues were promptly transferred to the ATL lysis buffer from QIAamp DNA Micro kit (Qiagen, 56304), and total DNA was extracted following the manufacturer's instructions. To detect paternal mtDNA in embryos, mass crosses were performed between 2-day-old $w^{1118}$ ($mt:ND2^{del1}$) females and 2-day-old $w^{1118}$ ($mt:wt$) or $poldip2^{del}$ ($mt:wt$) males. Embryos were collected for 30 min on standard grape juice agar plates, and total DNA was extracted from embryos using QIAamp DNA Micro kit at the indicated developmental time points.

Droplet digital PCR (ddPCR) was employed to quantify paternal mtDNA copy number in sperm and embryos. Due to the expected much higher abundance of mtDNA compared to single-copy nuclear genes, quantification using the duplex method was deemed unreliable. Therefore, a simplex ddPCR approach was used, where mtDNA and nuDNA assays were analyzed separately using different amounts of input DNA. To specifically detect paternal wild-type mtDNA ($mt: wt$) without detecting maternal mtDNA carrying the 9-bp deletion ($mt:ND2^{del1}$), a primer pair and a double-quenched FAM-labeled probe were designed. Additionally, the primers /probes targeting a Y-chromosome gene $kl-2$, and mtDNA-encoded $mt:CoI$ were used to quantify the sperm numbers in spermatheca and total mtDNA copy number in embryos, respectively. Primers/probe sets (IDT) used in the ddPCR reaction are listed in Table EV2.

Total DNA from *Drosophila* tissues was digested with EcoRI enzyme at 37 °C for 1 h. Subsequently, a ddPCR reaction mix containing 1× ddPCR Supermix for Probes (Bio-Rad, 186-3023), 250 nM of the probe, 900 nM of each primer, and the DNA template was assembled. The reactions were conducted in the QX200 ddPCR system (Bio-Rad), which includes droplet generation using QX200 droplet generator, PCR reactions on a C1000 Touch thermal cycler, and analysis on a droplet reader. The cycling conditions were as follows: For $mt:ND2$ detection, one cycle of 95 °C for 10 min, 42 cycles of 95 °C (2 °C/second ramp) for 30 s, 51 °C (2 °C/second ramp) for 1 min, 72 °C (2 °C/second ramp) for 15 s, one cycle of 98 °C for 10 min, 4 °C hold. For $mt:CoI$ and Y-chromosome gene $kl-2$, one cycle of 95 °C for 10 min, 40 cycles of 95 °C (2 °C /second ramp) for 30 s, 60 °C (2 °C/second ramp) for 1 min, one cycle of 98 °C for 10 min, 4 °C hold. The QuantaSoft analysis software (Bio-Rad) was used to acquire and analyze data.

To evaluate the specificity of the primers/probe set for $mt:ND2$, a reaction containing 10 ng of total DNA from $w^{1118}$ ($mt:ND2^{del1}$) flies mixed with varying amounts (0, 0.001, 0.005, 0.01, or 0.05 ng) of total DNA from $w^{1118}$ ($mt:wt$) flies was performed (Fig. EV2G,G'). The coefficient of correlation ($R^2$) was calculated to be 0.9947, indicating a good correlation between the input $w^{1118}$ ($mt:wt$) DNA amount and the measured copy number. It is noted that although the background signal is low, the input $w^{1118}$ ($mt:wt$) DNA amount lower than 0.001 ng is outside the linear range. Furthermore, the background signal arising from the presence of a large amount of $w^{1118}$ ($mt:ND2^{del1}$) DNA (from maternal mtDNA) was considered and subtracted in the calculation.

## Purification of Poldip2 protein

To produce recombinant Poldip2 protein, the *poldip2* coding sequence was cloned in frame with the C-terminal 6His tag of pET21b vector. The plasmid was transformed, and the protein was expressed in BL21(DE3) competent cells (Thermo Scientific, EC0114). Bacteria were cultured in Luria-Bertani medium at 37 °C until the optical density at the wavelength of 600 nm reached ~0.6. Subsequently, 0.4 mM isopropyl-β-ᴅ-thiogalactoside (IPTG) (Sigma) was added and the bacteria were cultured for an additional 20 h at 18 °C. Cells were harvested and lysed in buffer A (50 mM sodium phosphate, pH 7.4, 0.3 M NaCl, 10 mM imidazole, 5% (vol/vol) glycerol, 10 mM β-mercaptoethanol) supplemented with 1 mg/ml lysozyme (Sigma) and EDTA-free protease inhibitor (Roche, 11873580001) for 1 h on ice, followed by sonication five times for

5 min each. The cell lysates were clarified by centrifugation at $12,000 \times g$ for 30 min. The supernatants were first purified using affinity chromatography on a HisTrap column (Cytiva,17524802) on an ÄKTA pure protein purification system (Cytiva). The column was washed sequentially with 40 mM and 80 mM imidazole (Sigma) in buffer A, and the bound proteins were eluted with 250 mM imidazole in buffer A. The eluted protein was dialyzed against buffer B (20 mM Tris-HCl, pH 7.5, 0.1 M NaCl, 5% (v/v) glycerol, and 1 mM DTT) and then loaded onto a 5 ml HiTrap heparin column (Cytiva, 17040703) that had been equilibrated with buffer B. Following washing with buffer B, Poldip2 was eluted with a 40 ml gradient of 0.1 M to 1 M NaCl in buffer B, with Poldip2 eluting at salt concentrations of 0.1–0.3 M. The Poldip2-containing fraction was concentrated and further purified using a Superdex 200 increase 10/300 GL size-exclusion chromatography column (Cytiva, 28990944) equilibrated in buffer C (20 mM Tris-Cl, pH 7.5, 20 mM NaCl, 5% (v/v) glycerol, 1 mM DTT and 0.1 mM EDTA). The purified proteins were stored at $-80\,°C$ and the protein concentration was determined using Bradford plus protein assay reagents (Thermo Scientific, 23238).

## Nuclease assay

All 6-FAM-labeled DNA oligos were synthesized using the RNase-free HPLC purification method (IDT). To generate the dsDNA substrate, equal molar oligonucleotides were mixed in nuclease-free duplex buffer (IDT), heated to 95 °C for 5 min, and gradually cooled to 25 °C for 45 min. The nuclease assay was carried out using a reaction mixture containing 10 mM Tris-HCl, pH 8.3, 2.5 mM $MgCl_2$, 0.5 mM $CaCl_2$, 5 mM DTT, and 100 nM of 5'- or 3'-6-FAM-labeled ssDNA or dsDNA substrate (Table EV3). The reaction was initiated by adding the purified Poldip2 protein with the concentrations indicated in the figure legends, incubated at 37 °C for the indicated time and terminated by adding Novex™ TBE Urea sample buffer (Thermo Scientific, LC6876). The reaction mix was denatured for 5 min at 75 °C and resolved on a 20% denaturing polyacrylamide gel (SequaGel, National Diagnostics, EC-833) using a Model V16 vertical electrophoresis apparatus (15 cm × 17 cm × 0.8 mm, Apogee Electrophoresis) at 300 V for 2 h. The gels were imaged on a Typhoon biomolecular imager (GE Healthcare) using the fluorescence scanner (Cy2). The remaining substrate (%) was quantified using Fiji (NIH) and calculated by normalizing the substrate band density at each time point to that at time point zero.

## Transmission electron microscopy (TEM)

*Drosophila* testes were dissected in Schneider's *Drosophila* medium supplemented with 10% FBS and immediately fixed in a fixation solution (2.5% glutaraldehyde, 2% paraformaldehyde in 0.1 M sodium cacodylate buffer) at room temperature for 5 min, followed by an additional fixation on ice for 1 h. After washing in cold cacodylate buffer, the testes were postfixed with 2% Osmium tetroxide (Sigma), reduced with 1.5% potassium ferrocyanide immediately before use, for 1 h on ice. After washing with water, the tissues were placed in the thiocarbohydrazide (Electron Microscopy Sciences) solution for 20 min at room temperature. The testes were then fixed in 2% Osmium tetroxide for 30 min at room temperature, stained *en bloc* with 1% uranyl acetate (Electron Microscopy Sciences) overnight at 4 °C, and further stained with Walton's lead aspartate solution for 30 min at 60 °C. After

dehydration with ethanol series, the samples were embedded in Epon-Araldite (Electron Microscopy Sciences). The 80 nm thin sections, cut using a Leica EM UC6 ultramicrotome, were viewed on a Tecnai T12 (FEI, Hillsboro, OR) transmission electron microscope.

## TUNEL assay

*Drosophila* testes were dissected in Schneider's *Drosophila* medium supplemented with 10% FBS and fixed in PBS containing 4% paraformaldehyde for 20 min. After washing three times in PBS, the tissues were permeabilized in 0.25%Triton X-100 in PBS for 20 min. The testes were then processed following the instructions of the APO-BrdU kit (Phoenix Flow Systems, AU1001). Briefly, the samples were rinsed twice in the wash buffer before being incubated in the DNA-labeling solution containing Tdt enzyme, Br-dUTP, and Tdt reaction buffer. The reaction was carried out at 37 °C for 1 h, after which the samples were rinsed twice in rinse buffer. The samples were blocked in blocking buffer (2% BSA in PBS) for 60 min, then incubated with the anti-BrdU antibody (1:200, abcam, ab152095) for 2 h at room temperature or overnight at 4 °C. After three washes with PBS, the samples were incubated with Alexa Fluor 647 Phalloidin and Alexa Fluor 568 goat anti-rabbit IgG (1:200, Invitrogen). Finally, the samples were mounted in Vectashield antifade mounting medium with DAPI. Images were acquired using a PerkinElmer UltraView confocal system.

## DAPI staining of embryos

Crosses were performed between 2-day-old $w^{1118}$ females and either DJ-MTS-Red, *poldip2^{del}*, or DJ-MTS-Red control males. Embryos were collected 0–30 min post-laying on standard grape juice agar plates at 25 °C. The collected embryos were dechorionated using 50% bleach, followed by thorough rinsing with water. Fixation was performed in a 1:1 mixture of heptane and 4% formaldehyde in PBS for 25 min. After fixation, the formaldehyde layer was removed, and methanol was added to the remaining solution. The mixture was vigorously shaken for 15 s to devitellinize the embryos, with only those sinking to the bottom retained for further processing. The devitellinized embryos were washed three times with methanol and rehydrated by washing three times in PBTA solution (1% BSA, 0.05% Triton X-100 in PBS, pH 7.4). The embryos were subsequently stained with 1 μg/mL DAPI in PBTA solution for 5 min. Finally, the samples were mounted using Vectashield antifade mounting medium. Images were acquired using a Nikon CSU-W1 SoRa confocal system (Nikon SR Plan Apo IR 60×/1.27 oil lens; Nikon Element software; Yokogawa CSU-W1 SoRa Confocal Scanner Unit).

## Single-molecule fluorescence in situ hybridization (smFISH) of mtDNA

The labeling of mtDNA by TFAM-mNeonGreen in testes was assessed using single-molecule fluorescence in situ hybridization (smFISH) assay, following established protocols (Hurd et al, 2016). Briefly, testes from TFAM-mNeonGreen knock-in flies were dissected and fixed in a fixative buffer (100 mM sodium cacodylate buffer, pH 7.3, 100 mM sucrose, 40 mM potassium acetate, 10 mM sodium acetate, 10 mM EGTA, 5% paraformaldehyde) for 4 min.

Subsequently, the samples underwent sequential washing steps with $2 \times$ SSCT buffer ($2 \times$ SSC with 0.1% Tween-20), $2 \times$ SSCT/20% formamide, $2 \times$ SSCT/40% formamide, and $2 \times$ SSCT/50% formamide, each for 10 min. To make hybridization probes, 30 pairs of $5'$ labeled CAL Fluor Red 590 DNA oligonucleotide primers (Hurd et al, 2016) were synthesized (LGC Biosearch Technologies) and used to PCR amplify DNA fragments from $w^{1118}$ genomic DNA. The resulting ~300 bp PCR products from the 30 reactions were gel purified, pooled with equal molarity, and added to the hybridization solution ($2 \times$ SSC, 50% formamide, 10% dextran sulfate, 2 mg/ml BSA, 10 mM vanadyl ribonucleoside complex (Sigma, 94742)). Testes were denatured in the hybridization solution containing CAL Fluor Red 590 labeled probes (5 ng/µl) at 91 °C for 2 min, followed by overnight hybridization at 37 °C. The following day, samples were subjected to washing steps, starting with incubation in pre-warmed $2 \times$ SSCT/50% formamide solution at 37 °C, followed by room temperature incubation with $2 \times$ SSCT/40% formamide, $2 \times$ SSCT/20% formamide, and finally $2 \times$ SSCT. Lastly, the samples were mounted in Vectashield antifade mounting medium with DAPI, and images were acquired using a PerkinElmer UltraView confocal system.

### Sperm mitochondrial membrane potential staining

Seminal vesicles from male flies of the indicated genotypes were dissected in Schneider's *Drosophila* medium supplemented with 10% FBS and transferred to a slide with PBS containing 500 nM Tetramethylrhodamine (TMRM, Invitrogen, I34361) and 500 nM MitoTracker green (Invitrogen, M7514). After covering the tissue with a coverslip, sperm were extruded from seminal vesicles by gently applying pressure on the coverslip. Imaging was performed immediately to minimize the effects of hypoxia. Live images were captured using a Visitech instant structured illumination microscope (iSIM, BioVision) with 488–561 dual-camera acquisition mode (Olympus UPlanApo 60x/1.3 Silicone oil lens; Visiview acquisition software; ORCA-Flash4.0 V2 Digital CMOS camera C11440; excitation wavelength at 488 nm and 561 nm; exposure time 300 millisecond). To calculate the TMRM/MitoTracker Green ratios, image analysis was performed using Fiji (NIH). First, the 488 nm and 561 nm channels were separated in a single image. Then, regions of interest (ROI) within a sperm were selected on the TMRM channel (561 nm) using the "Color Threshold" function. The "Restore Selection" function was applied to outline the same area on the corresponding MitoTracker Green channel (488 nm). The mean intensity of the ROI and the background area in each channel were obtained through "mean gray value" using the "Measure" function. Finally, the intensity of TMRM and Mito-Tracker Green was obtained by subtracting the background intensity from the ROI intensity, respectively. The ratiometric values were generated by normalizing the mean intensity of TMRM to that of MitoTracker Green channels.

### Detection of ROS levels

Detection of ROS levels was performed using CellROX Deep Red according to the manufacturer's protocol (Thermo Scientific, C10422). Briefly, testes were incubated in Schneider's *Drosophila* medium containing 5 µM CellROX Deep Red for 45 min at 25 °C, followed by washing with PBS and fixation in 3.7% paraformaldehyde for 15 min. Samples were then mounted in Vectashield antifade mounting medium with DAPI. Images were captured using a PerkinElmer UltraView confocal system, and CellROX intensity was quantified using Fiji (NIH). Regions of interest (ROIs) within a seminal vesicle were selected on the CellROX Deep Red channel (640 nm). The mean intensity of the CellROX signal was obtained using the "Measure" function on the selected ROI, with background subtracted.

### Protein structure alignment

The protein structure of *Drosophila* Poldip2 was predicted in the AlphaFold protein structure database (AlphaFold DB: Q9VNC0). Pairwise Structure Alignment tool (https://www.rcsb.org/alignment) (Bittrich et al, 2024) was used to align two protein structures using the jFATCAT(rigid) alignment method.

### Statistical analyses

Data are shown as the mean ± SD (standard deviation). GraphPad Prism software 10.0 was used to generate charts and perform statistical analyses. Multiple unpaired $t$ tests were used to determine significant differences between two groups. The $P$ value is indicated by stars: ***$P < 0.0001$; **$P < 0.005$; *$P < 0.05$.

### Graphics

Figure EV5 graphics were created with BioRender.com.

## Data availability

The original images of this study have been deposited in the BioImage Archive with accession number S-BIAD1506. https://www.ebi.ac.uk/biostudies/bioimages/studies/S-BIAD1506. The source data of this paper are collected in the BioStudies database with accession number S-BSST1809. https://www.ebi.ac.uk/biostudies/studies/S-BSST1809.

The source data of this paper are collected in the following database record: biostudies:S-SCDT-10_1038-S44318-025-00377-5.

## Peer review information

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

## Acknowledgements

The authors thank Dr. S. Deluca for advice on the project; Dr. E Arama for sharing DJ-MTS-DsRed fly line; Dr. X Chen for proving FRT-SV40 PolyA-FRT pBluescript KS (−) plasmid; Bloomington Drosophila Stock Center; Drosophila Genomics Resource Center for various plasmids; BestGene for Drosophila injection service. This work is supported by the National Heart, Lung, and Blood Institute Intramural Research Program (1ZIAHL006153).

## Author contributions

**Zhe Chen**: Conceptualization; Data curation; Formal analysis; Validation; Investigation; Visualization; Methodology; Writing—original draft; Project administration; Writing—review and editing. **Fan Zhang**: Data curation; Formal analysis; Validation; Investigation; Visualization; Methodology. **Annie Lee**: Validation; Investigation; Visualization. **Michaela Yamine**: Validation; Investigation; Visualization. **Zong-Heng Wang**: Formal analysis; Validation; Investigation; Visualization. **Guofeng Zhang**: Formal analysis; Validation; Visualization. **Christian Combs**: Resources; Software; Supervision; Methodology. **Hong Xu**: Conceptualization; Resources; Supervision; Funding acquisition; Investigation; Writing—original draft; Project administration; Writing—review and editing.

Source data underlying figure panels in this paper may have individual authorship assigned. Where available, figure panel/source data authorship is listed in the following database record: biostudies:S-SCDT-10_1038-S44318-025-00377-5.

## Funding

## Disclosure and competing interests statement

The authors declare no competing interests.

# Expanded View Figures

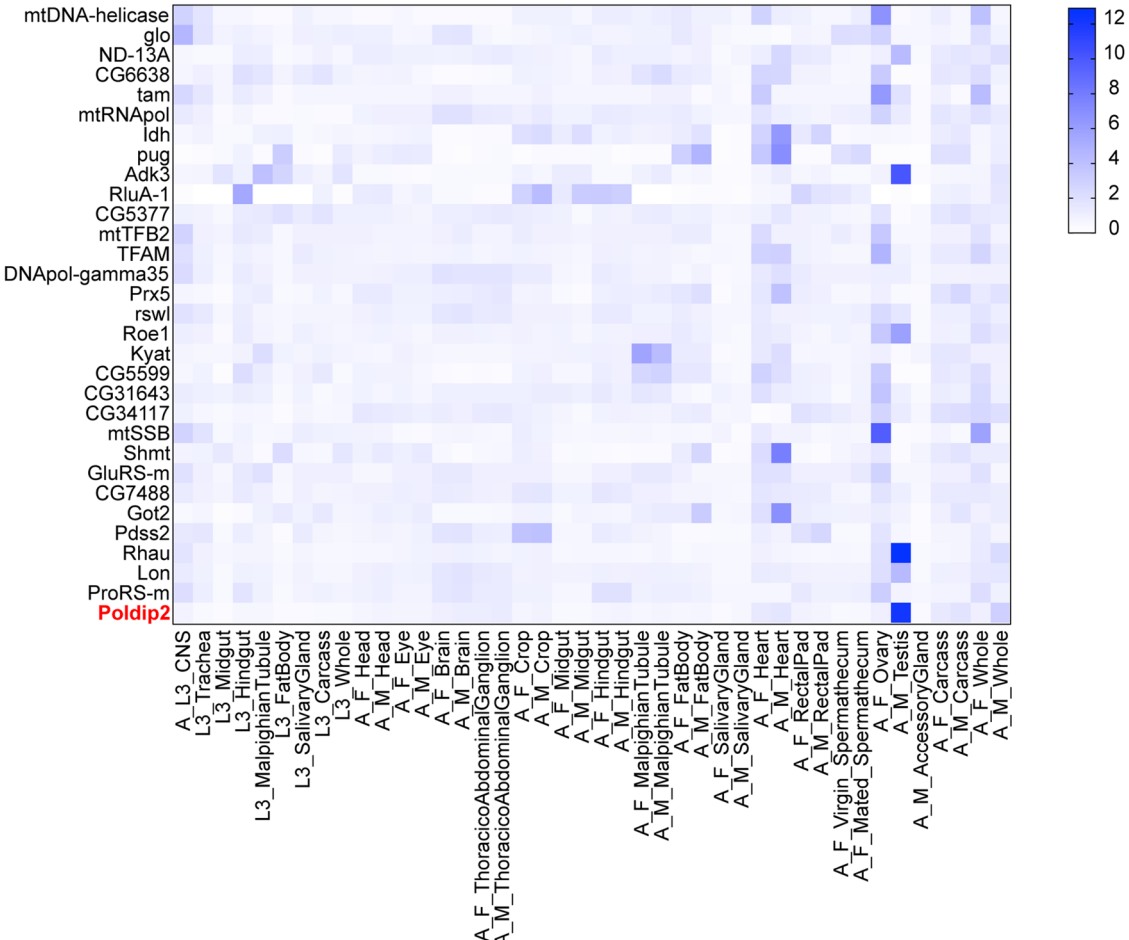

**Figure EV1. Tissue expression profile of *Drosophila* mitochondrial nucleoid-associated proteins.**

The heatmap was generated with RNA-seq data from FlyAtlas2 (the *Drosophila* gene expression atlas). Color codes indicate the scaled RPKM (reads per kilobase per million mapped reads) folds over tissues. Note that the mRNA level of Poldip2 is significantly higher in *Drosophila* testes compared to other tissues. A, adult; L3, third instar larva; F, female; M, male.

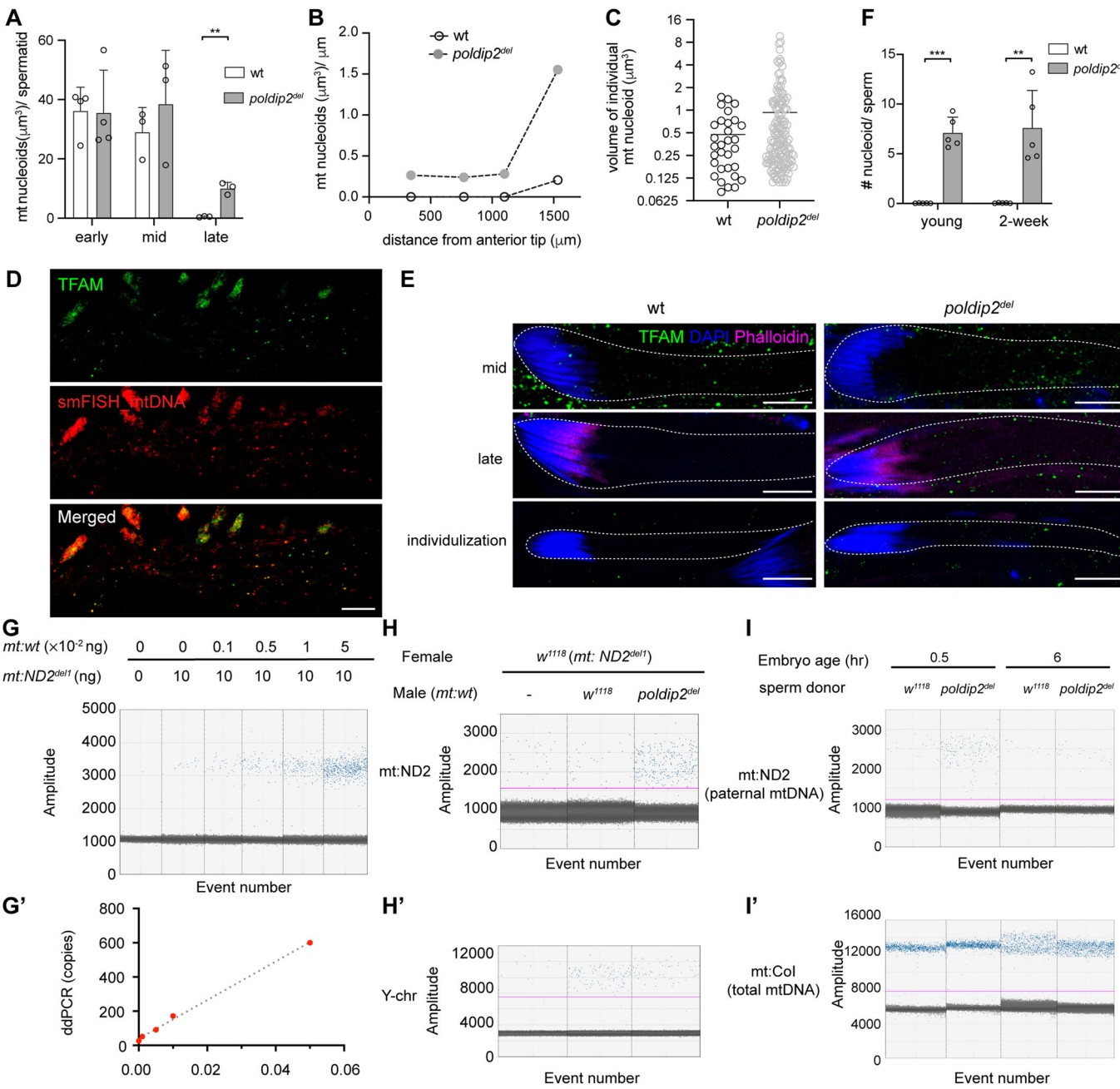

◀ **Figure EV2.   Mitochondrial DNA persists in late spermatogenesis stages and mature sperm of *poldip2^del* flies.**

(A) Total mitochondrial nucleoids measured in volumes per spermatid at early-elongating (early), mid-elongating (mid) and fully elongated (late) stages. Each data point represents a spermatid ($n = 3, 4$). The data represent the mean± SD. Statistical analysis was performed using an unpaired *t* test. **$P = 0.0012$. (B) The density of mitochondrial nucleoids (total volumes per µm) along the length of a representative fully elongated spermatid bundle for $w^{1118}$ (wt) and *poldip2^del* flies, respectively. (C) A scatter dot plot displaying the individual mitochondrial nucleoid volumes from the fully elongated spermatids of $w^{1118}$ (wt) and *poldip2^del* flies. Each data point represents an individual nucleoid (wt, $n = 31$; *poldip2^del*, $n = 153$). The solid lines indicate the mean volume. (D) TFAM-mNeonGreen can be used as a mitochondrial nucleoid marker in *Drosophila* testis. Single-molecule fluorescent in situ hybridization (smFISH) signals using fluorescently labeled DNA probes specific for mtDNA (red), are colocalized with TFAM-mNeonGreen in *Drosophila* testis. Bar, 10 µm. (E) Mitochondrial nucleoids labeled by TFAM-mNeonGreen demonstrate a consistent pattern of mtDNA elimination during spermatogenesis in both $w^{1118}$ (wt) and *poldip2^del* flies, compared with DNA dye (DAPI) staining. In elongating spermatids of both wt and *poldip2^del* testis, intense TFAM-mNeonGreen puncta signals were detected. The signals were rare in fully elongated and individualization stage spermatids of wt flies. Conversely, persistent mtDNA was frequently observed in the same stages of *poldip2^del* spermatids. Phalloidin (magenta) stains actin; DAPI (blue) stains nuDNA. Bar, 10 µm. (F) Quantification of mitochondrial nucleoid numbers in mature sperm of young and 2-week-old flies. Each data point represents quantification from one seminal vesicle ($n = 5$). The data represent the mean± SD. Statistical analysis was performed using an unpaired *t* test. ***$P = 7.0 \times 10^{-6}$, **$P = 0.0019$. (G) Evaluating the specificity of the primers/probe set targeting mtDNA-encoded ND2 locus (*mt:ND2*) using droplet digital PCR (ddPCR) assay. A reaction containing 10 ng of total DNA from $w^{1118}$ (*mt:ND2^del1*), a fly strain carrying a 9-base pair deletion on the mtDNA-encoded ND2 locus, mixed with 0, 0.001, 0.005, 0.01 or 0.05 ng of total DNA from $w^{1118}$ (*mt:wt*, wild-type mtDNA) flies, was performed. The ddPCR primers/probe were designed to target the *mt:wt* while excluding *mt:ND2^del1* mtDNA. (G') Correlation of the amount of input $w^{1118}$ (*mt:wt*) DNA with the resulting mtDNA copy numbers using ddPCR. Simple linear regression was carried out and the coefficient of correlation $R^2 = 0.9947$. (H–H') Quantification of mtDNA copy numbers per sperm in $w^{1118}$ and *poldip2^del* flies using ddPCR. Crosses were conducted between female $w^{1118}$ (*mt:ND2^del1*) and male $w^{1118}$ (*mt:wt*) or *poldip2^del* (*mt:wt*) flies. Then the total DNA from the female spermatheca was extracted and analyzed. Virgin female $w^{1118}$ (*mt:ND2^del1*) flies were used as the negative control. The primers/probe sets were designed to target *mt:ND2* (H) and Y-chromosome gene *kl-2* (H'), respectively. The input total DNA for detecting *mt:ND2* gene is 5 ng for each reaction. The input total DNA for detecting the Y-chromosome gene is 125 ng for each reaction. (I–I') Analysis of sperm-derived mtDNA in embryos using ddPCR. Crosses were performed between female $w^{1118}$ (*mt:ND2^del1*) and male $w^{1118}$ (*mt:wt*) or *poldip2^del* (*mt:wt*) flies. Embryos were collected 0–30 min post-laying and analyzed immediately (0.5 h) or after 6 h (6 h) of development. Primers/probe sets targeting *mt:ND2* (I) and *mt:CoI* (I') were used to quantify paternal mtDNA and total mtDNA, respectively. Total mtDNA levels in embryos remain constant up to 10 h after egg-laying (Rubenstein et al, 1977), averaging $2.5 \times 10^6$ copies per embryo (Appendix Fig. S1E). Input maternal mtDNA for detecting *mt:ND2* and *mt:CoI* was $2.5 \times 10^6$ and $2 \times 10^3$ copies per reaction, respectively.

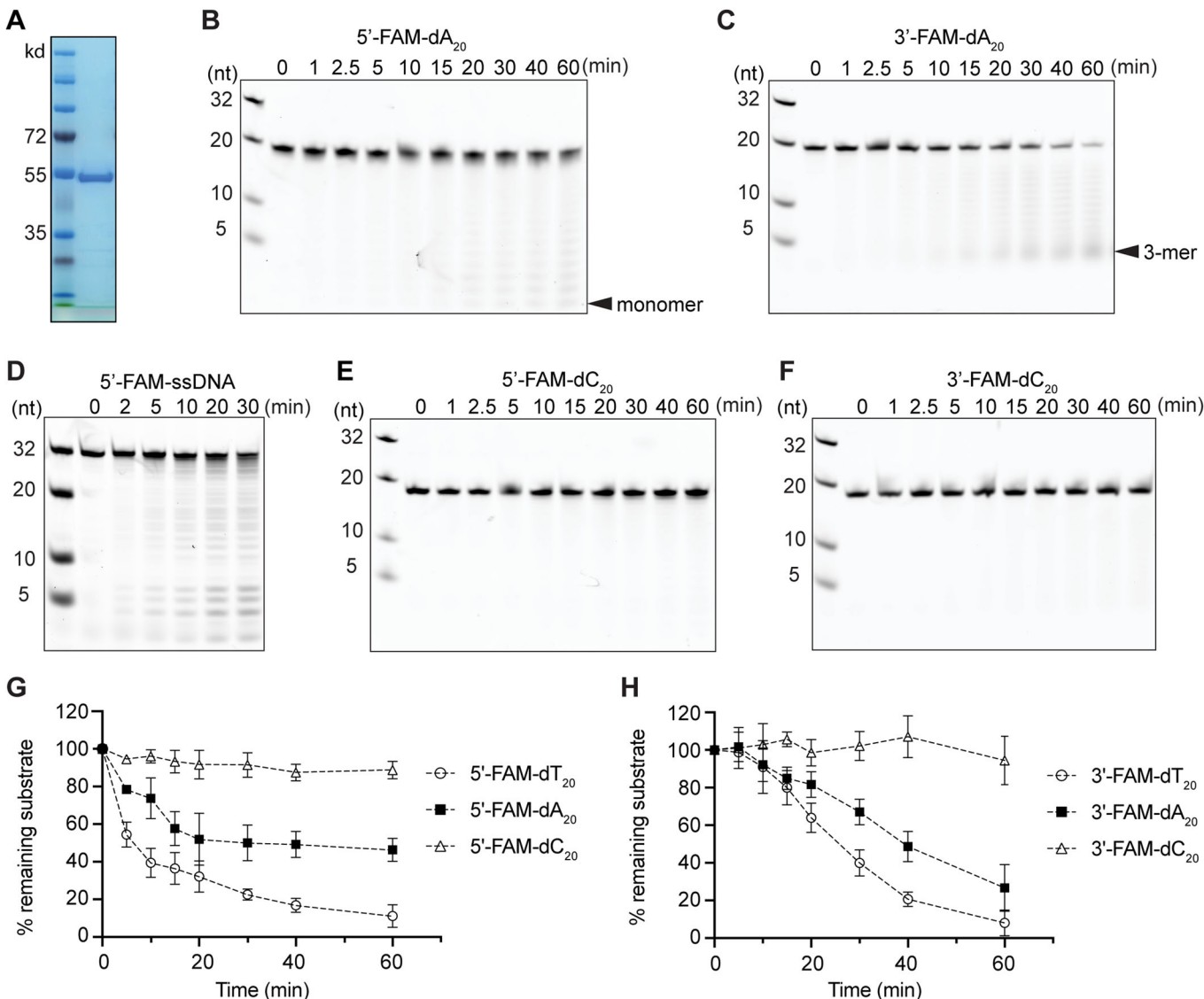

**Figure EV3. Poldip2 is a mitochondrial DNA exonuclease.**

(A) SDS-PAGE of the purified Poldip2 protein. M, molecular weight marker. (B, C) Degradation pattern of 5′-6-FAM and 3′-6-FAM-labeled 20-nt poly (dA) single-stranded (ssDNA) substrates. The 100 nM 5′-6-FAM (B) or 3′-6-FAM (C) labeled 20-nt poly (dA) was incubated with Poldip2 protein (200 nM) at 37 °C and analyzed at the indicated time points. (D) Degradation pattern of a 5′-6-FAM-labeled ssDNA substrate consisting of mixed dA, dT, dC and dG. The 100 nM 5′-6-FAM-labeled 32-nt ssDNA was incubated with Poldip2 protein (200 nM) at 37 °C and analyzed at the indicated time points. (E, F) Degradation pattern of 5′-6-FAM and 3′-6-FAM-labeled 20-nt poly (dC) ssDNA substrates. The 100 nM 5′-6-FAM (E) or 3′-6-FAM (F) labeled 20-nt poly(dC) was incubated with Poldip2 protein (200 nM) at 37 °C and analyzed at the indicated time points. (G, H) Quantification of the remaining full-length substrates, including 5′-6-FAM (G) or 3′-6-FAM (H) labeled 20-nt poly (dT), poly (dA) and poly (dC), at each time point. Data are normalized to the initial level of the full-length substrates and plotted ($n = 3$). The molecular markers in this figure are an equal molar mixture of 5′-6-FAM-labeled 32-nt, 20-nt, 10-nt and 5-nt oligonucleotides and were loaded at a concentration of 50 nM for each. The data represent the mean± SD.

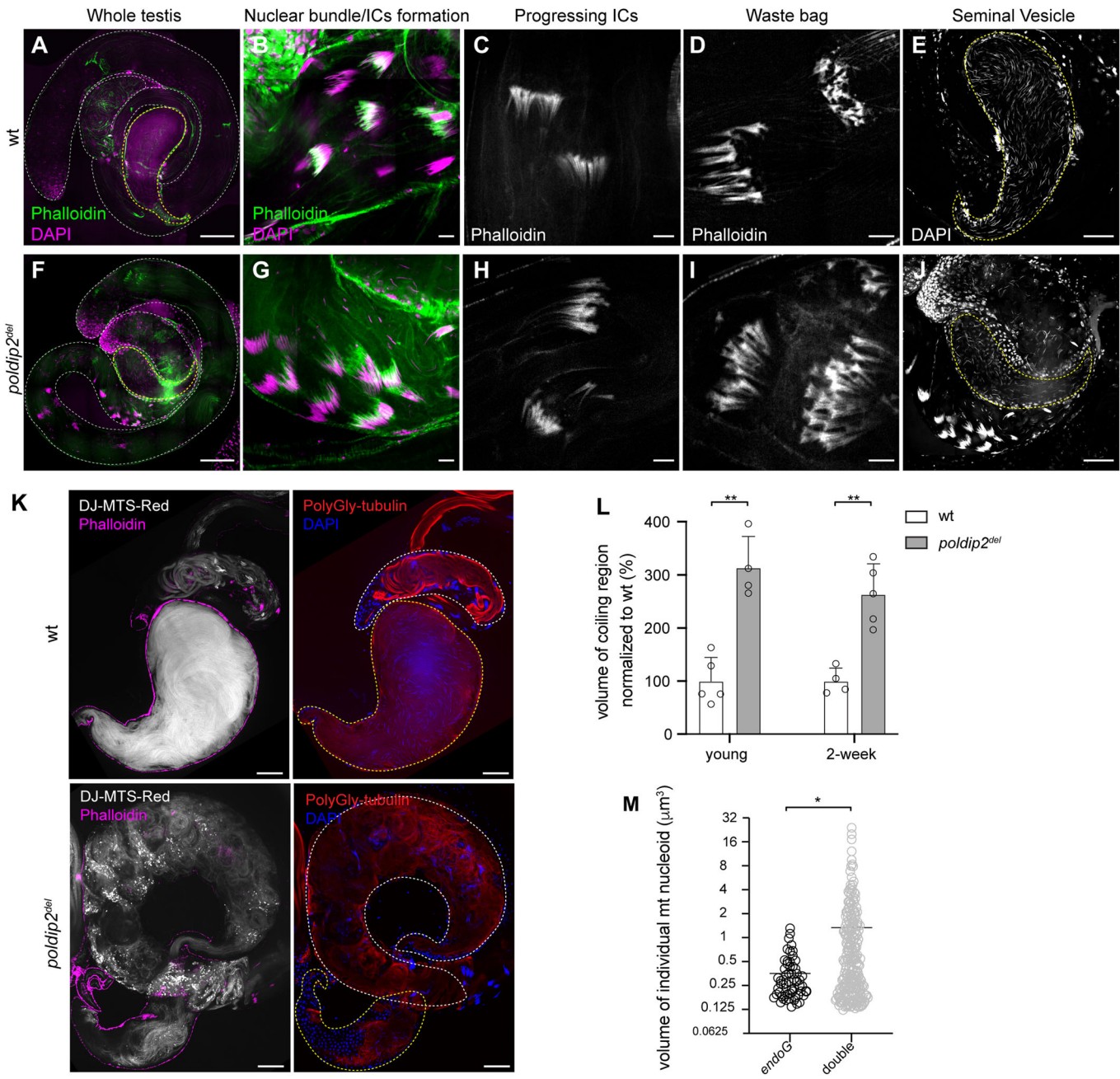

**Figure EV4. Persistent mtDNA impedes spermatid individualization.**

(A–J) Representative images showing the whole testis (**A**, **F**), individualization complexes (ICs) formation next to the nuclear head in the testis basal region (**B**, **G**), actin cone structures in progressing ICs (**C**, **H**), waste bags (**D**, **I**), and seminal vesicles (**E**, **J**, yellow dashed line) in *w^1118* (wt) and *poldip2^del* flies. Phalloidin stains actin; DAPI stains nuDNA. Bar, 100 μm in (**A**) and (**F**); 10 μm in (**B–D**) and (**G–I**); 50 μm in (**E**) and (**J**). (**K**) Representative images showing the coiling region (white dashed line) and seminal vesicle (yellow dashed line) of *w^1118* (wt) and *poldip2^del* flies. DJ-MTS-Red stains the mitochondria derivatives; Polyglycylated tubulin stains fully elongated axonemal microtubules; Phalloidin stains actin; DAPI stains nuDNA. Bar, 50 μm. (**L**) The coiling region in both young and 2-week-old *poldip2^del* flies is enlarged compared to wt control. Each data point represents quantification from one testis (*n* = 4, 5). The data represent the mean± SD. Statistical analysis was performed using an unpaired *t* test. *P* values from left to right: **\*\*P* = 0.00042, *P* = 0.0012. (**M**) A scatter dot plot displaying the individual mitochondrial nucleoid volumes from the elongated spermatids of *endoG* (*endoG^{MB07150/KO}*) and double mutants (*endoG^{MB07150/KO}*; *poldip2^del*). Each data point represents an individual nucleoid (*endoG*, *n* = 58; double mutant, *n* = 248). The solid lines indicate the mean volume. Statistical analysis was performed using an unpaired *t* test. *\*P* = 0.0058.

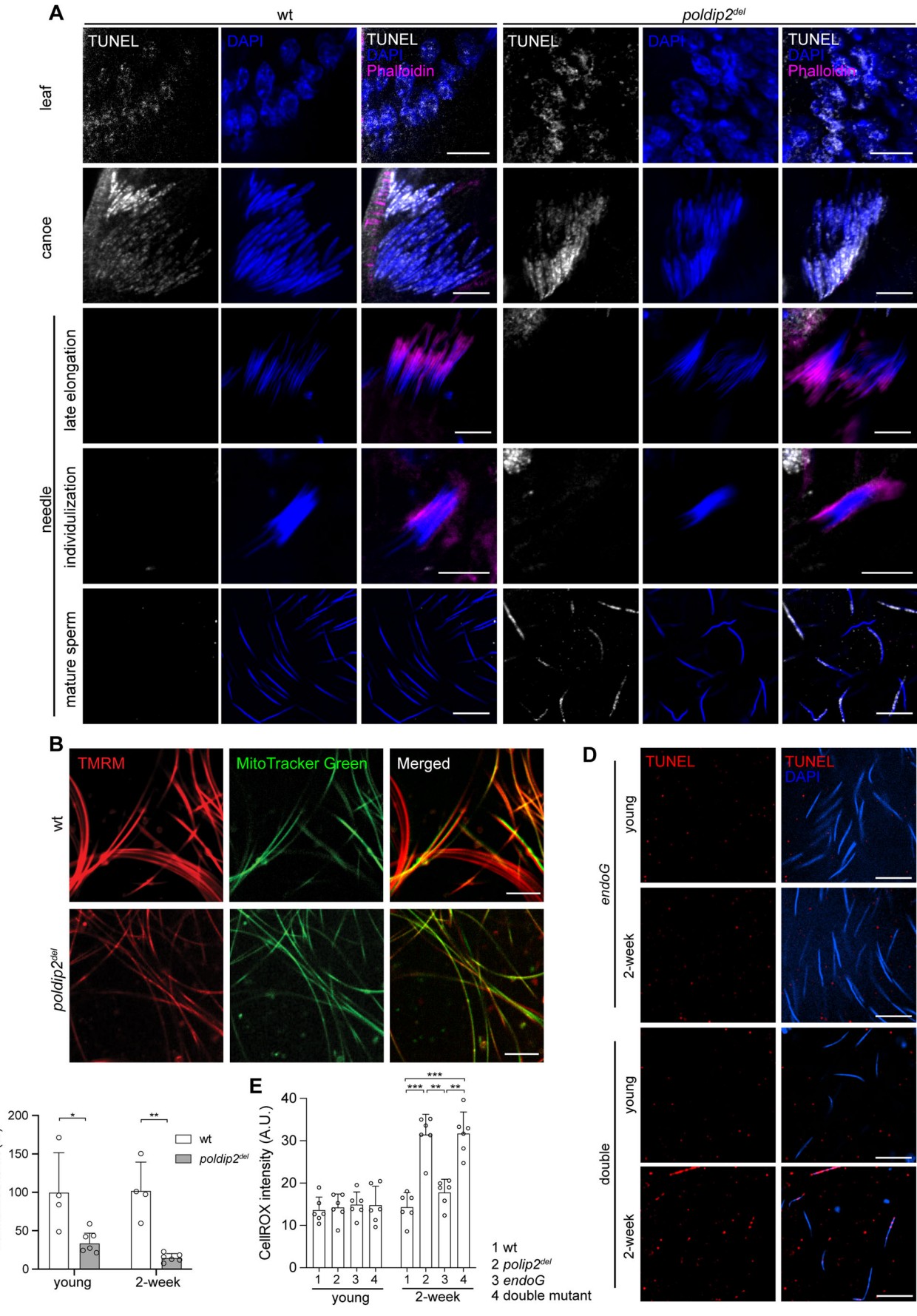

    

◀ **Figure EV5. Persistent mtDNA in mature sperm causes nuclear DNA fragmentation.**

(A) Representative images showing nuDNA breaks labeled by TUNEL assay in the process of chromatin remodeling during spermatogenesis. The highest abundance of TUNEL signal (white) was observed in the late canoe stage, which corresponds to the histone-to-protamine transition phase. The nuDNA breaks subsequently disappeared in needle-shaped nuclei during late elongation and individualization stages, indicating the repair of nuDNA breaks after the transition. No significant differences were observed between wt and *poldip2^del* flies throughout this process. The developmental stages were distinguished by the morphology of nuclear heads stained with DAPI (blue), and the positioning of actin cones stained with Phalloidin (magenta). Bar, 10 μm. (B) Compromised mitochondrial membrane potential of *poldip2^del* sperm. Mature sperm from *w^1118* (wt) and *poldip2^del* seminal vesicles were stained with TMRM (red), a dye sensitive to mitochondrial membrane potential, in combination with MitoTracker Green (green) as a reference. Bar, 10 μm. (C) Quantification of TMRM/MitoTracker Green ratios in both young and 2-week-old flies. Each data point represents quantification from one seminal vesicle (*n* = 4, 6). The data represent the mean± SD. Statistical analysis was performed using an unpaired *t* test. \*P = 0.019, \*\*P = 0.00039. (D) Representative images of TUNEL assay in *endoG* (*endoG^MB07150/KO*) and double mutant (*endoG^MB07150/KO*; *poldip2^del*) seminal vesicles. The nuDNA breaks/fragmentation was observed in 2-week-old double mutant mature sperm. DAPI (blue) stains the nuDNA of mature sperm. Red: TUNEL signal. Bar, 10 μm.

(E) Quantification of CellROX intensity, as a measure of ROS levels in both young and 2-week-old flies with indicated genotypes. (1) *endoG^MB07150*/+; *poldip2 ^del*/+. (2) *endoG^MB07150*/+; *poldip2 ^del*. (3) *endoG^MB07150/KO*; *poldip2 ^del*/+. (4) *endoG^MB07150/KO*; *poldip2^del*. A.U., arbitrary unit. Each data point represents quantification from one seminal vesicle (*n* = 6). The data represent the mean± SD. Statistical analysis was performed using an unpaired *t* test. *P* values from left to right: \*\*\*P = 0.000038, \*\*\*P = 0.000032, \*\*P = 0.00016, \*\*P = 0.00018.

 