## [Peer Review File · The EMBO Journal]

Mitochondrial DNA removal is essential for sperm development and activity

Zhe Chen, Fan Zhang, Annie Lee, Michaela Yamine, Zong-Heng Wang, Guofeng Zhang, Christian Combs and Hong Xu

Corresponding author: Hong Xu (hong.xu@nih.gov)

Review Timeline:

Submission Date:	3rd Jun 24
Editorial Decision:	5th Aug 24
Revision Received:	3rd Dec 24
Editorial Decision:	8th Jan 25
Revision Received:	22nd Jan 25
Accepted:	24th Jan 25

Editor: Ieva Gailite

Transaction Report:

Dear Hong and Zhe,

Thank you for submitting your manuscript for consideration by the EMBO Journal. I sincerely apologise for the protracted assessment process due to delays in referee report submission. We have now received comments from three reviewers, which are included below for your information.

As you can see, all reviewers are generally positive in their assessment and find the proposed role of Poldip2/ExoA novel and of interest to the readership of our journal. However, they also indicate several concerns that would need to be addressed before they can support publication. From the nomenclature aspect, reviewers #1 and #3 find renaming of Poldip2 to ExoA unnecessary, and I agree that this might cause confusion. From my side, I find the raised points generally reasonable. I would therefore invite you to address these remaining comments in a revised manuscript. I think that it would be useful to discuss the revision in more detail via email or phone/videoconferencing - please let me know which option you prefer.

We generally allow three months as standard revision time. Should you foresee a problem in meeting this deadline, please let us know in advance to discuss an extension. As a matter of policy, competing manuscripts published during this period will not negatively impact on our assessment of the conceptual advance presented by your study. However, please contact me as soon as possible upon publication of any related work to discuss the appropriate course of action.

When preparing your letter of response to the referees' comments, please bear in mind that this will form part of the Review Process File and will therefore be available online to the community. For more details on our Transparent Editorial Process, please visit our website: <https://www.embopress.org/page/journal/14602075/authorguide#transparentprocess>. Please also see the attached instructions for further guidelines on preparation of the revised manuscript.

Please feel free to contact me if have any further questions regarding the revision. Thank you for the opportunity to consider your work for publication, and I look forward to discussing your revision with you.

With best regards,

leva

leva Gailite, PhD
Senior Scientific Editor
The EMBO Journal
Meyerhofstrasse 1
D-69117 Heidelberg
Tel: +4962218891309
i.gailite@embojournal.org

- a point-by-point response to the referees' comments, with a detailed description of the changes made (as a word file).
- a word file of the manuscript text.
- individual production quality figure files (one file per figure)
- a complete author checklist, which you can download from our author guidelines (<https://www.embopress.org/page/journal/14602075/authorguide>).

- Expanded View files (replacing Supplementary Information)

- a Reagents and Tools Table as part of the Methods section, which can be downloaded from our author guidelines

(<https://www.embopress.org/page/journal/14602075/authorguide#structuredmethods>)

We realize that it is difficult to revise to a specific deadline. In the interest of protecting the conceptual advance provided by the work, we recommend a revision within 3 months (3rd Nov 2024). Please discuss the revision progress ahead of this time with the editor if you require more time to complete the revisions.

Referee #1:

This study by Chen et al. and the related study by Wang et al. identify a new protein, Poldip2, that contributes to developmentally-programmed mtDNA elimination during spermatogenesis. Both studies are important because they extend our mechanistic understanding of how mtDNA is strictly maternally inherited in animals. Chen et al. additionally shows that failure to eliminate mtDNA induces multiple developmental and physiological defects in sperm that compromise fertility. Chen et al. is particularly interesting because it addresses why mtDNA elimination during spermatogenesis is such a highly conserved process in many animals.

The following conclusions in Chen et al are well supported by the data:

- 1) Poldip2 localizes to sperm and S2 cell mitochondria.
- 2) Poldip2 expression coincides with mtDNA elimination during sperm development.
- 3) Poldip2 mutants partially compromise mtDNA elimination during spermatogenesis.
- 4) Poldip2 mutants are semi-sterile, and both their sterility and mtDNA elimination phenotypes can be rescued by a bacterial exonuclease artificially-targeted to sperm mitochondria.
- 5) Purified Poldip2 has exonuclease activity in vitro. Together with finding #4, these elegant experiments argue that Poldip2 is a mitochondrial exonuclease, and that Poldip2's exonuclease activity promotes fertility through mtDNA elimination.
- 6) Poldip2 mutants have fewer mature sperm than WT or mtExoIII-rescued controls. Poldip2 mutant sperm also produce more ROS and have more damaged nuclei than these controls. These experiments suggest a model where sperm mtDNA elimination enhances sperm quality by reducing ROS which can damage nuclear DNA.

There are no major problems with this paper and in my opinion, it is suitable for publication in your journal with only minor revisions.

Minor Criticisms:

- 1) The authors propose renaming Poldip2 to "ExoA." It is acceptable to rename a gene but only if the new name is favored by the research community. In this case, it would be prudent to have both Chen et al. and Wang et al. agree on the ExoA name (or something else) before publication.
- 2) The authors show that Poldip2 expression coincides with mtDNA elimination during spermatogenesis. This begs the question: is Poldip2 expression sufficient to deplete mtDNA? The authors expressed Poldip2-mCherry in S2 cells. Did these S2 cells have reduced mtDNA?
- 3) The authors depiction of the Poldip2 phenotype in the "coiling region" is difficult to see in figure 4a. The authors might consider using DJ-GFP sperm to better visualize the coiling sperm bundles.
- 4) In the EM in figure 4C, the authors do not comment on the lack of white substance (glycogen?) in the Poldip2 sperm that normally forms next to the major mitochondrial derivative. See left panels top vs bottom. In the bottom right panel, the black arrowhead looks like its pointing toward residual cytoplasm, not a mitochondrial derivative (dark circle to the right of the flagella). Clearly the individualized Poldip2 sperm look different than controls though.
- 5) The authors propose a reasonable model where defective mtDNA elimination compromises mitochondrial membrane potential, leading to ROS production that induces DNA damage. While further testing this model is outside of the scope of the paper, the authors should note that reduced mitochondrial membrane potential could have other effects on cell physiology that could lead to nuclear DNA damage (for example, failure to import or retain mitochondrial EndoG).

6) The main difference between Chen et al. and Wang et al. is that Wang et al detected paternal mtDNA leakage into the adult offspring of Poldip2 mutant fathers. Chen et al. concluded that "Paternal mtDNA was detected in embryos 30 minutes after egg laying but disappeared 6 hours later (Extended Data Fig. 2h, h'), consistent with the notion that sperm mitochondria are destroyed during early embryogenesis." The raw ddPCR data in Extended figure 2h is not easy for the reader to interpret and translate into this conclusion. Furthermore, this conclusion is likely unfounded with the authors' experimental design. The authors apparently loaded the same total DNA into each ddPCR. However, after 6 hours of development, paternal mtDNA will be heavily diluted by exponentially-replicating nuclear DNA. In other words, even if there were the same number of paternal mtDNA genomes per embryo at 0.5h and 6 hours, the authors would have loaded far less paternal mtDNAs into the 6 hour ddPCR. For this reason, the authors should either remove the 6-hour timepoint or try other normalization methods (like normalizing to # embryos or maternal mtDNA copy number).

7) Poldip2-derived embryos had considerably fewer Y chromosomes than controls (figE2h') at both 0.5 and 6 hours. This result suggests that either a large proportion of Poldip2 embryos are not fertilized, or that they fail early development and do not replicate their DNA. The authors should either comment on these possibilities or discriminate between them by imaging 1-3 hour-old embryos with DJ-GFP (to measure fertilization) and DAPI (to measure developmental progression).

Referee #2:

In this paper, Chen et al, found Poldip2 is involved in paternal mtDNA clearance in sperm and normal spermatogenesis in *D. melanogaster*. They found that Poldip2 is exclusively enriched in the testis and is colocalized with mitochondrial nucleoid. In the Poldip2 KO mutant fly, mtDNA clearance during spermatogenesis is impaired. Poldip2 KO also showed defects in sperm individualization and reduced male fertility. They demonstrated that recombinant Poldip2 protein has exonuclease activity in vitro. Furthermore, expression of bacterial exonuclease in sperm can partially rescue the reduced fertility of poldip2 KO, suggesting that Poldip2 functions as an exonuclease.

This paper provides mechanistic insights into the clearance of sperm mtDNA and suggests its relationship to normal sperm development. The paper is well-organized and would interest the readership of EMBO Journal. However, before publication, the following points must be addressed.

Major comments

1. Poldip2 was originally identified as a human polymerase P50-interacting protein and has been proposed to be involved in nuclear genome replication and repair. In this study, the authors showed that a recombinant Poldip2 protein has an exonuclease activity in vitro. However, we cannot exclude the possibility that some bacterial protein co-purified with Poldip2 shows exonuclease activity. Some negative controls should be included to conclude that Poldip2 is a novel exonuclease. It would be essential to examine point mutants of active sites of the catalytic domain or conserved residues (e.g. DNA-binding defective YvvC domain mutants).

2. Does Poldip2 show a similarity to any known nucleases such as an *E.coli* Exonuclease III? Which domain of Poldip2 is responsible for the exonuclease activity? It would be helpful to compare their 3D structures using Aphafold2. Some explanation or discussion should be included as well.

3. The authors concluded that persistent mtDNA impedes individualization. However, the EndoG mutant showed a defect in mtDNA clearance but its fertility is normal, suggesting that individualization defect is not directly correlated to the level of remaining mtDNA. Since Poldip2 mutant also showed abnormal mtDNA nucleoid organization. In addition, reduced mitochondrial membrane potential was observed, suggesting that mitochondrial activity is impaired. Therefore, we cannot exclude the possibility that such phenotypes might be more directly related to individualization defects. At least, they should mention this possibility.

4. The localization and expression pattern of Poldip2 are important points. Please confirm that ExoA-mCherry (Fig. 1a) and ExoA-mNeonGreen (Fig. 1f) are functional. In addition, it would be better to show that ExoA-mNeonGreen is concentrated in mtDNA nucleoids in sperm, too.

5. It has been reported that rab7 and atg7 are involved in the degradation of paternal mitochondria in embryos. Do these mutations (rab7 or atg7) enhance the paternal mtDNA leakage phenotype of Poldip2 mutants?

Minor comment

1. The authors renamed Poldip2 to ExoA. However, it might be more confusing if Poldip2 has no sequence similarity to the known "ExoA" family.

2. p1, line 13-14

"During multicellular organisms' reproduction, organelles, cytoplasmic materials, and mitochondrial DNA (mtDNA) are all derived from maternal lineage."

Since centriole is provided from sperm in many species, "all" is inappropriate.

3. p2 line 7,

"However, increasing evidence suggests that mtDNA and mitochondria-derived structures are actively eliminated during spermatogenesis or embryogenesis, respectively."

"mitochondria-derived structures" may be confusing. One might think as mitochondrial-derived vesicles.

4. p3 line 32

"Additionally. ExoA was found" should be "Additionally, ExoA was found".

5. p5 line 6

"E. coli" should be "Escherichia coli".

6. Fig.4c

It is difficult to see the black arrow.

7. Fig.4 e-f

For direct comparison, it would be better to include images of the ExoA single mutant.

Referee #3:

Summary and significance:

Chen et al. report the characterization of Poldip2 as an exonuclease that is expressed in spermatogenesis and contributes to the elimination of mtDNA during spermatogenesis of *Drosophila melanogaster*. Poldip2 mutant males retain abnormally high levels of mtDNA in mature sperm and are able to transmit their mtDNA into *Drosophila* embryos, which is degraded during later embryo development. They show in vitro, that Poldip2 has exonuclease activity and their data supports the hypothesis that Poldip2 works in concert with the previously described endonuclease endoG. Additionally, the authors show that sperm maturation is blocked in Poldip2 mutants leading to male sterility, a phenotype that is progressing with male age.

We consider this study as an interesting and important body of work. The experiments are well designed and the data is presented in a clear way. The experiment showing that a bacterial exonuclease can rescue defects associated with Poldip2 loss of function is particularly nice and supports the requirement of Poldip2 exonuclease activity for mtDNA removal. The identification of factors contributing to mtDNA clearance in *Drosophila* sperm adds an important piece to the puzzle of safeguarding maternal inheritance of mitochondrial DNA.

Specific concerns:

We see no reason to rename the gene. Poldip2 is named similarly in other species and renaming it in flies would add unnecessary confusion and could reduce broader recognition of this work.

The statement that "persistent mtDNA in mature sperm causes marked fragmentation of the nuclear genome" (see abstract) is too strong. The causative link between these two observations is not identified and should thus be explored deeper or the claim should be removed from the text. For example, the TUNEL assay is insufficient to explain the observed sterility, and needs further experimental evidence or taken out of the manuscript.

Is the nuclear DNA also fragmented in an endoG/ Poldip2 mutant?

Are the measured CellROX levels increased in the endoG/ Poldip2 mutant as well. These data should be shown.

Do two-week-old flies retain more mtDNA in their testis and transmit to progeny compared to young males?

Minor points:

Fig 1F: Please show single channel images separately in addition to the overlay image provided to enable an assessment of ExoA expression timing in the testis.

The 3D projections in figure 2F (bottom) do not add any valuable information

The manuscript has frequent syntactic and grammatical errors (e.g. missing articles) that reduce readability of the text, which the authors should address before publication.

Referee #1:

This study by Chen et al. and the related study by Wang et al. identify a new protein, Poldip2, that contributes to developmentally-programmed mtDNA elimination during spermatogenesis. Both studies are important because they extend our mechanistic understanding of how mtDNA is strictly maternally inherited in animals. Chen et al. additionally shows that failure to eliminate mtDNA induces multiple developmental and physiological defects in sperm that compromise fertility. Chen et al. is particularly interesting because it addresses why mtDNA elimination during spermatogenesis is such a highly conserved process in many animals.

The following conclusions in Chen et al are well supported by the data:

- 1) Poldip2 localizes to sperm and S2 cell mitochondria.
- 2) Poldip2 expression coincides with mtDNA elimination during sperm development.
- 3) Poldip2 mutants partially compromise mtDNA elimination during spermatogenesis.
- 4) Poldip2 mutants are semi-sterile, and both their sterility and mtDNA elimination phenotypes can be rescued by a bacterial exonuclease artificially-targeted to sperm mitochondria.
- 5) Purified Poldip2 has exonuclease activity in vitro. Together with finding #4, these elegant experiments argue that Poldip2 is a mitochondrial exonuclease, and that Poldip2's exonuclease activity promotes fertility through mtDNA elimination.
- 6) Poldip2 mutants have fewer mature sperm than WT or mtExoIII-rescued controls. Poldip2 mutant sperm also produce more ROS and have more damaged nuclei than these controls. These experiments suggest a model where sperm mtDNA elimination enhances sperm quality by reducing ROS which can damage nuclear DNA.

There are no major problems with this paper and in my opinion, it is suitable for publication in your journal with only minor revisions.

Minor Criticisms:

- 1) The authors propose renaming Poldip2 to "ExoA." It is acceptable to rename a gene but only if the new name is favored by the research community. In this case, it would be prudent to have both Chen et al. and Wang et al. agree on the ExoA name (or something else) before publication.

We appreciate the reviewer's comment. We considered the name "Poldip2" misleading, as the protein's mitochondrial localization makes a functional interaction with Polymerase delta, a nuclear protein, unlikely. Nonetheless, we understand the importance of community consensus in renaming genes. We will keep the original name, "Poldip2," in this manuscript to avoid potential confusion and ensure clarity for readers.

2) The authors show that Poldip2 expression coincides with mtDNA elimination during spermatogenesis. This begs the question: is Poldip2 expression sufficient to deplete mtDNA? The authors expressed Poldip2-mCherry in S2 cells. Did these S2 cells have reduced mtDNA?

In the revised manuscript, we demonstrate that Poldip2 overexpression reduces mtDNA content in cultured S2 cells and *Drosophila* ovarian germ cells (Appendix Fig. S2A, B). However, we observed variable effects in terminally differentiated midgut enterocytes (ECs) (Appendix Fig. S2C). We hypothesize that Poldip2-mediated mtDNA degradation may depend on the presence of DNA breaks generated during active mtDNA replication, as observed in S2 cells and ovarian germ cells. These observations highlight a potential role of Poldip2 in mtDNA homeostasis. We included these data in Appendix Fig. S2 and revised the manuscript accordingly (page 6, line 23-31).

3) The authors depiction of the Poldip2 phenotype in the "coiling region" is difficult to see in figure 4a. The authors might consider using DJ-GFP sperm to better visualize the coiling sperm bundles.

To better visualize spermatid bundles in coiling region and mature sperms in seminal vesicles, we combined *Poldip2^{del}* lines with DJ-MTS-DsRFP transgene (Politi *et al*, 2014), which labels mitochondria derivatives in late spermatogenesis stages. Additionally, we co-stained spermatid bundles and sperms with polyglycylated tubulin, which marks fully elongated axonemal microtubules. These images have been included in Fig. EV4K in the revised manuscript.

4) In the EM in figure 4C, the authors do not comment on the lack of white substance (glycogen?) in the Poldip2 sperm that normally forms next to the major mitochondrial derivative. See left panels top vs bottom. In the bottom right panel, the black arrowhead looks like its pointing toward residual cytoplasm, not

a mitochondrial derivative (dark circle to the right of the flagella). Clearly the individualized *Poldip2* sperm look different than controls though.

Currently, the nature of the white substance observed in control sperms, and the reason for its absence in *Poldip2*^{del} sperms, remain unclear to us. The arrowhead in the bottom-right panel is intended to indicate two spermatids that have failed individualization and are still connected to each other. We have adjusted these arrowheads in the figure to improve clarity.

5) The authors propose a reasonable model where defective mtDNA elimination compromises mitochondrial membrane potential, leading to ROS production that induces DNA damage. While further testing this model is outside of the scope of the paper, the authors should note that reduced mitochondrial membrane potential could have other effects on cell physiology that could lead to nuclear DNA damage (for example, failure to import or retain mitochondrial EndoG).

We agree with the reviewer that a potential causative link between oxidative stress and the nuclear genome fragmentation remains to be established (page 9, lines 4 to 10). We do not favor the idea that EndoG might contribute to the nuclear genome fragmentation, as the EndoG/*Poldip2* double mutant, which had high level of residual mtDNA in mature sperm, also displayed marked nuclear genome fragmentation (Fig. EV5D). We include this result in the revised manuscript (page 8, lines 22-28).

6) The main difference between Chen et al. and Wang et al. is that Wang et al. detected paternal mtDNA leakage into the adult offspring of *Poldip2* mutant fathers. Chen et al. concluded that "Paternal mtDNA was detected in embryos 30 minutes after egg laying but disappeared 6 hours later (Extended Data Fig. 2h, h'), consistent with the notion that sperm mitochondria are destroyed during early embryogenesis." The raw ddPCR data in Extended figure 2h is not easy for the reader to interpret and translate into this conclusion. Furthermore, this conclusion is likely unfounded with the authors' experimental design. The authors apparently loaded the same total DNA into each ddPCR. However, after 6 hours of development, paternal mtDNA will be heavily diluted by exponentially replicating nuclear DNA. In other words, even if there were the same number of paternal mtDNA genomes per embryo at 0.5h and 6 hours, the authors would have loaded far less paternal mtDNAs into the 6 hour ddPCR. For this reason, the

authors should either remove the 6-hour timepoint or try other normalization methods (like normalizing to # embryos or maternal mtDNA copy number).

We greatly appreciate the reviewer's insight on this issue. A previous study (Rubenstein *et al*, 1977) reported that total embryonic mtDNA content remains unchanged up to ten hours after egg laying. We hence used maternal mtDNA copy number as a reference to normalize paternal mtDNA at different time points. Our data support this observation and further demonstrate that mtDNA content remains constant within our experimental time window, irrespective of whether the embryos are fertilized or not (Appendix Fig. S1E).

The new results show that paternal mtDNA from *poldip2^{del}* sperm was detectable in embryos 30 minutes after egg laying but disappeared by 6 hours (Fig. 2H, Fig. EV2I, I'). In addition, following a suggestion from another reviewer, we maternally knocked down ATG7, a key enzyme involved in initiating autophagy pathway, in embryos. The knockdown resulted in higher levels of paternal mtDNA in early embryos, although it was eventually eliminated (Appendix Fig. S1D). These results are consistent with the previous study showing that sperm mitochondria are degraded through multiple pathway during early embryogenesis (Politi *et al.*, 2014). The above results are described on page 5, lines 4-8 in the revised manuscript.

7) Poldip2-derived embryos had considerably fewer Y chromosomes than controls (figE2h') at both 0.5 and 6 hours. This result suggests that either a large proportion of Poldip2 embryos are not fertilized, or that they fail early development and do not replicate their DNA. The authors should either comment on these possibilities or discriminate between them by imaging 1-3 hour-old embryos with DJ-GFP (to measure fertilization) and DAPI (to measure developmental progression).

We greatly appreciate the reviewer's suggestions. In the revised manuscript, we assessed the success of fertilization by visualizing DJ-MTS-DsRed that marks sperm tail in embryos (Politi *et al.*, 2014) and examined embryonic development using DAPI staining to indicate nuclear cycles. In the *poldip2^{del}* group, fewer embryos were fertilized, and fewer embryos developed, both compared to wt group (Appendix Fig. S1A-C). Notably, embryonic development was initiated in all fertilized embryos (containing sperm tail) in both groups (Appendix Fig. S1B), indicating that the reduced number of developing embryos results from the reduced fertilization in the *poldip2^{del}* group. These observations also suggest that

the reduced success of fertilization in the *poldip2^{del}* group is likely caused by the lower quality and quantity of sperm produced by *poldip2^{del}* flies. The above results are described on page 5, lines 1-4 in the revised manuscript.

Referee #2:

In this paper, Chen et al, found Poldip2 is involved in paternal mtDNA clearance in sperm and normal spermatogenesis in *D. melanogaster*. They found that Poldip2 is exclusively enriched in the testis and is colocalized with mitochondrial nucleoid. In the Poldip2 KO mutant fly, mtDNA clearance during spermatogenesis is impaired. Poldip2 KO also showed defects in sperm individualization and reduced male fertility. They demonstrated that recombinant Poldip2 protein has exonuclease activity in vitro. Furthermore, expression of bacterial exonuclease in sperm can partially rescue the reduced fertility of *poldip2* KO, suggesting that Poldip2 functions as an exonuclease. This paper provides mechanistic insights into the clearance of sperm mtDNA and suggests its relationship to normal sperm development. The paper is well-organized and would interest the readership of EMBO Journal. However, before publication, the following points must be addressed.

Major comments

1. Poldip2 was originally identified as a human polymerase δ P50-interacting protein and has been proposed to be involved in nuclear genome replication and repair. In this study, the authors showed that a recombinant Poldip2 protein has an exonuclease activity in vitro. However, we cannot exclude the possibility that some bacterial protein co-purified with Poldip2 shows exonuclease activity. Some negative controls should be included to conclude that Poldip2 is a novel exonuclease. It would be essential to examine point mutants of active sites of the catalytic domain or conserved residues (e.g. DNA-binding defective YvC domain mutants).

Based on the predicted structure of *Drosophila melanogaster* Poldip2 (AlphaFold DB: Q9VNC0) and amino acids conservation across orthologs, we constructed five mutants within YvC domain. Unfortunately, none of these mutants could be stably expressed in *E. Coli*, preventing us from further assessing their activity within the limited timeframe of this revision.

We surveyed all DNA exonucleases currently identified in *E. coli* (Lovett, 2011) and found that none exhibits the enzyme properties of recombinant Poldip2,

which degrades both ssDNA and dsDNA, in both directions, and prefers A/T over G/C. While we cannot entirely rule out the possibility of bacterial protein contamination, we consider this scenario unlikely.

2. Does Poldip2 show a similarity to any known nucleases such as an E.coli Exonuclease III? Which domain of Poldip2 is responsible for the exonuclease activity? It would be helpful to compare their 3D structures using AlphaFold2. Some explanation or discussion should be included as well.

Poldip2 protein is primarily consisted of two domains: a YccV domain that potentially binds to hemimethylated DNA, and an ApaG domain, whose function remains unknown. The protein structure of *Drosophila melanogaster* Poldip2 has been predicted in AlphaFold Protein Structure Database (AlphaFold DB: Q9VNC0). Using the Pairwise Structure Alignment tool (Bittrich *et al*, 2024), we found that Poldip2's structure differs from most common nuclease structures (Yang, 2011), including *E. coli* Exonuclease III. Interestingly, Poldip2 shares a moderate structural similarity with mammalian Dom3Z (TM-score 0.21), a recently identified RNA nuclease with 5'-3' exoribonuclease activity (Jiao *et al*, 2013). The C-terminal ApaG domain appears to be the region with the most structural resemblance, suggesting it may be related to the nuclease activity. We have added the data in Appendix Fig. S3 and the description on page 6 lines 4-5, and page 9, lines 20-22.

3. The authors concluded that persistent mtDNA impedes individualization. However, the EndoG mutant showed a defect in mtDNA clearance, but its fertility is normal, suggesting that individualization defect is not directly correlated to the level of remaining mtDNA. Since Poldip2 mutant also showed abnormal mtDNA nucleoid organization. In addition, reduced mitochondrial membrane potential was observed, suggesting that mitochondrial activity is impaired. Therefore, we cannot exclude the possibility that such phenotypes might be more directly related to individualization defects. At least, they should mention this possibility.

We would like to clarify the differences in both timing and outcomes of mtDNA elimination defects between the *endoG* mutant and *Poldip2^{del}* flies. In the *endoG* mutant, although mtDNA clearance is delayed in elongated spermatids, mtDNA is ultimately cleared before the individualization stage, and the mutant flies have normal fertility. In contrast, in *Poldip2^{del}* fly, mtDNA persists beyond the individualization stage and remains in mature sperms, resulting in semi-sterility.

Therefore, the phenotype observed in the *endoG* mutant does not contradict to our proposal that persistent mtDNA impedes individualization.

In *poldip2^{del}* flies, we observed multiple mitochondrial defects, including persistent mtDNA, impaired mitochondrial membrane potential, and increased ROS levels. These defects were largely restored by expressing a mitochondrially-targeted *E. coli* ExoIII, which ectopically degrades persistent mtDNA in mutant sperm. Thus, we consider defective mtDNA clearance to be the root cause underlying other mitochondrial abnormalities.

Nonetheless, we agree with the reviewer that other mitochondrial defects, such as impaired mitochondrial activity, may also contribute to the individualization defects and reduced fertility in young mutant flies. We have clarified this point in the revised manuscript (page 11, lines 3-4).

4. The localization and expression pattern of Poldip2 are important points. Please confirm that ExoA-mCherry (Fig. 1a) and ExoA-mNeonGreen (Fig. 1f) are functional. In addition, it would be better to show that ExoA-mNeonGreen is concentrated in mtDNA nucleoids in sperm, too.

The Poldip2-mNeonGreen transgene was generated by inserting the mNeonGreen cDNA in-frame into the endogenous locus of *polidp2* gene. Homozygous male transgenic flies, in which both copies are tagged with mNeonGreen, were healthy overall and particularly, displayed normal fertility compared to wt males, suggesting that Poldip2-mNeonGreen is functional. We have clarified this point on page 18, lines 23-24.

Additionally, we demonstrated that Poldip2-mCherry reduced mtDNA levels when ectopically expressed in S2 cells (Appendix Fig. S2A) and ovarian germ cells (Appendix Fig. S2B), further corroborating its functionality.

The expression of Poldip2-mNeonGreen begins when mtDNA clearance occurs (Fig.1F). Therefore, we are unable to demonstrate co-localization of Poldip2-mNeonGreen with mtDNA nucleoids in sperm, as the presence of Poldip2 coincides with the absence of mtDNA nucleoids.

5. It has been reported that *rab7* and *atg7* are involved in the degradation of paternal mitochondria in embryos. Do these mutations (*rab7* or *atg7*) enhance

the paternal mtDNA leakage phenotype of Poldip2 mutants?

We appreciate the reviewer's suggestion. In the revised manuscript, ATG7 was maternally knocked down in *mt:ND2^{del1}* embryos using a maternal triple driver (*MTD-gal4*) according to a previous publication (Politi *et al.*, 2014). Using ddPCR assay, we did observe a higher amount of *poldip2^{del}* sperm mtDNA in the ATG7 RNAi group in early embryogenesis (both 30 min and 1 hour after egg laying), compared to control group. However, the paternal mtDNA was still eliminated after 3 hours, These results are consistent with the previous study showing that sperm mitochondria are degraded through multiple pathway during early embryogenesis (Politi *et al.*, 2014). We have included this data in Appendix Fig. S1D and described it on page 5, lines 4-8 in the revised manuscript.

Ectopically expressing the Rab7 dominant negative mutant (BDSC, 9778) in embryos using *matalpha4-gal4* (BDSC, 7063) driver caused developmental arrest in most embryos, preventing us from testing a potential involvement of Rab7 over the course of embryonic development.

Minor comment

1. The authors renamed Poldip2 to ExoA. However, it might be more confusing if Poldip2 has no sequence similarity to the known "ExoA" family.

We have reverted the name back to "Poldip2" to avoid potential confusion.

2. p1, line 13-14

"During multicellular organisms' reproduction, organelles, cytoplasmic materials, and mitochondrial DNA (mtDNA) are all derived from maternal lineage."

Since centriole is provided from sperm in many species, "all" is inappropriate.

This is a moot point now, as we have removed this sentence from the Abstract, due to the word limit.

3. p2 line 7,

"However, increasing evidence suggests that mtDNA and mitochondria-derived structures are actively eliminated during spermatogenesis or embryogenesis, respectively."

"mitochondria-derived structures" may be confusing. One might think as mitochondrial-derived vesicles.

We have changed “mitochondria-derived structures” to “mitochondrial-derived vesicles” in the revised manuscript (page 2, line 8).

4. p3 line 32

"Additionally. ExoA was found" should be "Additionally, ExoA was found".

We have corrected it.

5. p5 line 6

"E. coli" should be "Escherichia coli".

We have corrected it.

6. Fig.4c

It is difficult to see the black arrow.

We have changed the arrow color to yellow in Fig.4C.

7. Fig.4 e-f

For direct comparison, it would be better to include images of the ExoA single mutant.

Images of *poldip2^{del}* (single mutant) and wild-type control are shown in Fig.2A-C. They are not included in Fig.4 due to the space constraint.

Referee #3:

Summary and significance:

Chen et al. report the characterization of Poldip2 as an exonuclease that is expressed in spermatogenesis and contributes to the elimination of mtDNA during spermatogenesis of *Drosophila melanogaster*. Poldip2 mutant males retain abnormally high levels of mtDNA in mature sperm and are able to transmit their mtDNA into *Drosophila* embryos, which is degraded during later embryo development. They show in vitro, that Poldip2 has exonuclease activity and their data supports the hypothesis that Poldip2 works in concert with the previously described endonuclease endoG. Additionally, the authors show that sperm maturation is blocked in Poldip2 mutants leading to male sterility, a phenotype

that is progressing with male age.

We consider this study as an interesting and important body of work. The experiments are well designed and the data is presented in a clear way. The experiment showing that a bacterial exonuclease can rescue defects associated with Poldip2 loss of function is particularly nice and supports the requirement of Poldip2 exonuclease activity for mtDNA removal. The identification of factors contributing to mtDNA clearance in *Drosophila* sperm adds an important piece to the puzzle of safeguarding maternal inheritance of mitochondrial DNA.

Specific concerns:

We see no reason to rename the gene. Poldip2 is named similarly in other species and renaming it in flies would add unnecessary confusion and could reduce broader recognition of this work.

We have reverted the name back to “Poldip2” to avoid potential confusion.

The statement that "persistent mtDNA in mature sperm causes marked fragmentation of the nuclear genome" (see abstract) is too strong. The causative link between these two observations is not identified and should thus be explored deeper or the claim should be removed from the text. For example, the TUNEL assay is insufficient to explain the observed sterility, and needs further experimental evidence or taken out of the manuscript.

Ectopic expression of mitochondrially-targeted *E. coli* ExoIII, which removes residual mtDNA in *poldip2^{del}* sperm, but unlikely affecting other mitochondrial processes, largely rescues all other phenotypes, including nuclear genome fragmentation. These results support that persistent mtDNA is the root cause of the observed defects, although the intermediate processes leading to nuclear genome fragmentation remains to be elucidated. Nonetheless, we think the data of nuclear genome fragmentation, providing a potential explanation for the complete sterility of old male flies, is important.

We do agree with the reviewer that the causative link between the nuclear genome fragmentation and observed sterility has yet to be established. We have revised the manuscript to clarify this point (page 8, lines 14-16; page 9, lines 5-11).

Is the nuclear DNA also fragmented in an *endoG*/ *Poldip2* mutant?

We examined nuclear DNA fragmentation in the *endoG/Poldip2^{del}* double mutant using the TUNEL assay. Despite very few mature sperm present in seminal vesicles, we observed fragmented nuclear DNA in the 2-week-old double mutant. These data are included in Fig. EV5D with the description on page 8, lines 22-28 of the revised manuscript.

Are the measured CellROX levels increased in the *endoG*/ *Poldip2* mutant as well. These data should be shown.

The ROS level was increased in 2-week-old *endoG/Poldip2^{del}* double mutant. The data is shown in Fig. EV5E and page 9, lines 4-6 in the revised manuscript.

Do two-week-old flies retain more mtDNA in their testis and transmit to progeny compared to young males?

The number of mitochondrial nucleoids per sperm in seminal vesicles of 2-week-old *poldip2^{del}* flies is comparable to those in younger *poldip2^{del}* flies (Fig. EV2F). Presumably, spermatids with excessive mitochondrial nucleoids are too compromised to fully mature and consequently fail to be released into seminal vesicles. This may lead to an accumulation or "jam" in the coiled region of the testis, as shown in Fig. 4A and Fig. EV4K, ultimately disrupting spermatogenesis.

The 2-week-old *poldip2^{del}* flies become completely sterile, preventing us from examining the transmission of mtDNA to progeny at this age.

Minor points:

Fig 1F: Please show single channel images separately in addition to the overlay image provided to enable an assessment of ExoA expression timing in the testis.

We have added the single channel image of *Poldip2*-mNeonGreen in the revised Fig. 1F.

The 3D projections in figure 2F (bottom) do not add any valuable information

We have deleted the 3D projections in Fig.2F.

The manuscript has frequent syntactic and grammatical errors (e.g. missing

articles) that reduce readability of the text, which the authors should address before publication.

We have thoroughly reviewed the manuscript and addressed the grammatical and syntactic errors to improve the clarity and readability of the text.

Reference

- Bittrich S, Segura J, Duarte JM, Burley SK, Rose Y (2024) RCSB protein Data Bank: exploring protein 3D similarities via comprehensive structural alignments. *Bioinformatics* 40
- Jiao X, Chang JH, Kilic T, Tong L, Kiledjian M (2013) A mammalian pre-mRNA 5' end capping quality control mechanism and an unexpected link of capping to pre-mRNA processing. *Mol Cell* 50: 104-115
- Lovett ST (2011) The DNA Exonucleases of Escherichia coli. *EcoSal Plus* 4
- Politi Y, Gal L, Kalifa Y, Ravid L, Elazar Z, Arama E (2014) Paternal Mitochondrial Destruction after Fertilization Is Mediated by a Common Endocytic and Autophagic Pathway in Drosophila. *Developmental Cell* 29: 305-320
- Rubenstein JL, Brutlag D, Clayton DA (1977) The mitochondrial DNA of Drosophila melanogaster exists in two distinct and stable superhelical forms. *Cell* 12: 471-482
- Yang W (2011) Nucleases: diversity of structure, function and mechanism. *Q Rev Biophys* 44: 1-93

Dear Hong,

Thank you for submitting a revised version of your manuscript. I sincerely apologise for the delay in communicating the decision due to the holiday period.

We have now received input from two of the original reviewers, who find that most of previous concerns have been addressed satisfactorily. In a final minor revision, please tone down the conclusions as requested by reviewer #3. Please also add in the manuscript text a caveat for possible bacterial protein contamination in response to the comment by reviewer #2.

Additionally, there remain a few editorial points that need addressing before I can extend official acceptance of the manuscript:

1. Please submit up to five keywords.
2. Please make sure that the order of the sections in the manuscript is as follows: Abstract / Keywords / Introduction / Results / Discussion / Methods / Acknowledgments / Disclosure and competing interests statement / References / Figure legends / Tables and their legends / Expanded View Figure legends.
3. Please add headings "Abstract" and "Introduction" in the manuscript text and correct the heading "Material and Methods" to "Methods".
4. Please upload the Reagents and Tools Table as a separate file choosing the file type "Reagent Table".
5. Please remove the BioRender reference from figure legend and add a separate "Graphics" section to "Methods" as below:
Graphics:
Figure EV5 graphics were created with BioRender.com.
6. Please rename Table EV1 into Dataset EV1 and upload as a dataset file. Please add a legend to the file, in a separate tab/worksheet. Please renumber Tables EV2-4 accordingly and upload as "Expanded View Files". These tables also need legends added to the top of each table. Legends should be removed from the manuscript text. All callouts in the manuscript text will need to be updated accordingly.
7. Please rename "Competing interests" section into "Disclosure and competing interests statement" (further info: <https://www.embopress.org/page/journal/14602075/authorguide#conflictsofinterest>).
8. CRediT has replaced the traditional author contributions section because it offers a systematic, machine-readable author contributions format that allows for more effective research assessment. Please remove the Authors Contributions from the manuscript and use the free text boxes beneath each contributing author's name in our online submission system to add specific details on the author's contribution. More information is available in our guide to authors.
9. Please add the grant number to the "Acknowledgments" section in the manuscript text.
10. Please move "Data Availability" section to the end of "Methods" and add a resolvable URL to the BioImage Archive entry. Further information about the format of this section can be found here: <https://www.embopress.org/page/journal/14602075/authorguide#dataavailability>
11. Our data editors have flagged the following issues in figure legends that need correcting:
 - Please provide the exact p values in the legends of figures 1E, 2D, G, H; 3B, C; 4B, D, H; 5B, D; EV2 A, F; EV4 L, M; EV5 C, E.
 - Please define the measure of center for the error bars in the legends of figures 1E, 2D, G, H; 3B, C; 4B, D, H; 5B, D; EV2 A, F; EV3 G, H; EV4 L; EV5 C, E.
12. Papers published in The EMBO Journal are accompanied online by a 'Synopsis' to enhance discoverability of the manuscript. It consists of A) a short (1-2 sentences) summary of the findings and their significance, B) 3-4 bullet points highlighting key results and C) a synopsis image that is 550x300-600 pixels large (width x height, jpeg or png format). You can either show a model or key data in the synopsis image. Please note that the image size is rather small and that text needs to be readable at the final size. Please send us this information together with the revised manuscript.

With best wishes,

Ieva

Ieva Gailite, PhD
Senior Scientific Editor
The EMBO Journal
Meyerhofstrasse 1
D-69117 Heidelberg

Tel: +4962218891309
i.gailite@embojournal.org

We realize that it is difficult to revise to a specific deadline. In the interest of protecting the conceptual advance provided by the work, we recommend a revision within 3 months (8th Apr 2025). Please discuss the revision progress ahead of this time with the editor if you require more time to complete the revisions.

Referee #2:

In the revised manuscript, the authors have addressed most of my concerns. However, I still have concerns regarding the following points

In their response to the major comment 1, the authors stated that they have tried to express several point mutants of Poldip2 in *E. coli*, none of which were successful. We acknowledge their efforts and understand that some proteins are difficult to express in *E. coli*. However, since this in vitro enzyme assay is one of the main conclusions of this paper, I think that it is very important and essential to include appropriate negative controls.

Referee #3:

We read the revised version of the manuscript and consider it much improved. The authors responded to our and the other reviewer's comments and provided additional evidence. However, we still consider that a direct connection between nuclear DNA fragmentation and mitochondrial persistence has not been clearly established. In particular, the statements on page 9, line 2 'and further substantiate a causative role of persistent mtDNA in nuclear genome fragmentation.' and lines 13-15 'These findings suggest that residual mtDNA may impair mitochondrial respiration, leading to increased oxidative stress over time, which could contribute to nuclear genome damage' seem overinterpreted given the data shown.

The authors addressed the remaining editorial issues.

Dear Hong,

Thank you for submitting the final revised version and addressing the remaining editorial points. I am now pleased to inform you that your manuscript has been accepted for publication. Congratulations on a great study!

Before we forward your manuscript to our publishers, I would like to propose some edits in the manuscript abstract and synopsis (please also see the attached file). I have also written a short blurb that will accompany the title of your manuscript in our online system. Please take a look and let me know if any corrections are needed. Additionally, I noticed that the provided synopsis contains a couple of typos - the text in the left lower corner should say "healthy mature sperm devoid of mtDNA". Please send me a corrected file via email.

Blurb:

Identification of Poldip2 as exonuclease required for mtDNA clearance allows assessing the importance of this process during late stages of *Drosophila* spermatogenesis.

Synopsis:

Mitochondrial DNA (mtDNA) is actively removed during spermatogenesis in a conserved process that enforces the uniparental inheritance of mtDNA. Here, identification of Poldip2 as exonuclease essential for mtDNA elimination allows assessing the importance of this process for ensuring proper spermatogenesis and sperm functionality in *Drosophila*.

- Poldip2 is a mitochondrial exonuclease specifically expressed in late stages of *Drosophila* spermatogenesis.
- Loss of Poldip2 disrupts mtDNA clearance in elongated spermatids, impedes sperm individualization, and causes mtDNA to persist in mature sperm.
- Over time, persistent mtDNA in mature sperm of poldip2 mutant flies leads to an increased level of reactive oxygen species, nuclear genome fragmentation, and complete sterility.

If you have any questions, please do not hesitate to contact the Editorial Office. Thank you for this contribution to The EMBO Journal and congratulations on a nice study!

With best wishes,

Ieva
